# A new lidar design for operational atmospheric wind and cloud / aerosol survey from space

Didier Bruneau and Jacques Pelon

LATMOS/IPSL, Sorbonne Université, UVSQ, CNRS, Paris, France

*Correspondence to*: J. Pelon (jacques.pelon@latmos.ipsl.fr)

**Abstract.** Global wind profile measurement has for long been a first priority for numerical weather prediction. The demonstration from ground-based observations that a double-edge Fabry-Perot interferometer could be efficiently used for deriving wind profiles from the molecular scattered signal in a very large atmospheric vertical domain has led to the choice of

the direct detection technique in space and the selection of the Atmospheric Dynamic Mission (ADM) Aeolus by ESA in 1999. ADM-Aeolus was successfully launched in 2018, after the technical issues raised for the lidar development had been solved, providing first global wind profiles from space in the whole troposphere. Simulated and real time assimilation of the projected horizontal wind information were able to confirm the expected improvements in forecast score, validating the concept of a wind profiler using a single line-of-sight lidar from space.

The question is raised here about consolidating results gained from ADM-Aeolus mission with a potential operational follow-on instrument. Maintaining the configuration of the instrument as close as possible to the one achieved (UV emission lidar with a single line-of-sight) we revisit the concept of the receiver by replacing the arrangement of the Fizeau and Fabry-Perot interferometers with a unique Quadri-channel Mach-Zehnder (QMZ) interferometer which relaxes the system operational constraints and extends the observation capabilities to recover the radiative properties of clouds. This ability to profile wind

and cloud/aerosol radiative properties enables meeting two highest priorities of the meteorological forecasting community regarding atmospheric dynamics and radiation.

We discuss the optimization of the key parameters necessary in the selection of a high performance system, as based on previous work and development of our airborne QMZ lidar. The selected optical path difference (3.2 cm) of the QMZ leads to a very compact design allowing the realization of a high quality interferometer and offering a large field-angle acceptance.

Performance simulation of horizontal wind speed measurements with different backscatter profiles shows results in agreement with the targeted ADM-Aeolus random errors, using an optimal 45° line-of-sight angle. The Doppler measurement is, from principle, unbiased by the atmospheric conditions (temperature, pressure, particle scattering) and only weakly affected by the instrument calibration errors. The study of the errors arising from the uncertainties in the instrumental calibration and in the modelled atmospheric parameters used for the backscattered signal analysis shows a limited impact under realistic conditions.

The particle backscatter coefficients can be retrieved with uncertainties better than a few percent when the scattering ratio

exceeds 2 such as in the boundary layer and in semi-transparent clouds. Extinction coefficients can be derived accordingly. The chosen design further allows addition of a dedicated channel for aerosol and cloud polarization analysis.

## 1     Introduction

Direct wind profiles in the meteorological atmosphere (0-25 km) are lacking over the oceans and in the tropics as indirect retrievals from temperature sounding are of no help in this region due to the lack of geostrophic equilibrium. First priority in global atmospheric observations from space for weather forecasting was set on wind profiling to get better information on atmospheric circulation (Baker et al., 1995, 2014, WMO, 1996, 2012, Stoffelen et al., 2005, 2020).

Aerosols are good tracers of atmospheric dynamics as their reduced Brownian motion allows to perform high accuracy spectral measurements from the analysis of small Doppler shifts induced in lidar backscattered light. This is why the choice of heterodyne lidar was first targeted for space missions (LAWS Instrument Panel report, 1987, Baker et al., 1995). The feasibility of such heterodyne systems operating on particle scattering was discussed for long in the community, but an important drawback was that the vertical profiling extent was limited by the very low value of the backscatter coefficient of the upper tropospheric particles in the infrared. This leads to a very constraining lidar design in terms of mass and power needs for such a space operating system. Such studies were continued to examine solutions and contributions of a space-based Doppler lidar possibly operating in direct detection (Baker et al., 2014).

Lidar backscattered signal from molecules, though much more spectrally broadened due to their high speed thermal motion at molecular scale, was shown to be very effective for deriving upper tropospheric and stratospheric winds using direct detection. This was evidenced from pioneering work performed at Service d'Aéronomie -now Laboratoire ATmosphères, Milieux et Observations Spatiales -LATMOS- (Chanin et al., 1989, Garnier et al., 1992). Specific interferometric techniques involving multi Fabry-Perot in differential detection, also known as double-edge technique, were implemented to analyze Doppler shift due to mean atmospheric motions. A few drawbacks are inherent to this approach, such as to require a narrow field of view and a particulate scattering correction in the analysis to reduce biases (Garnier et al., 1990). For the first lidar space mission proposed to ESA, wavelength operation was selected in the ultraviolet (UV) because of eye-safety issues, and because it maximized molecular return. Though first observational tests in the UV from space revealed their difficulty (McCormick et al., 1993), the candidate Atmospheric Dynamics Mission was accepted as the first Earth Explorer mission by ESA in 1999. A technique combining two interferometers in cascade, one matched to the narrow aerosol spectrum (Mie channel), the other matched to the broad molecular spectrum (Rayleigh channel) was chosen to be implemented in the space-borne Atmospheric LAser Doppler INstrument (ALADIN) providing high signal to noise over the whole atmosphere. An airborne demonstrator was developed and showed that the required performance could be achieved (Reitebuch et al, 2009, Lux et al., 2020). The satellite launch occurred in 2018, after a rather long delay induced by the large number of problems to be overcome. Though recent data analyses (Witschas et al., 2020, Martin et al, 2021) show that the horizontal line-of-sight (HLOS) wind measurements exhibit seasonal and orbital dependent biases slightly larger than the missions requirements, the measurements

were confirmed to be highly beneficial and the European Centre for Medium-Range Weather Forecasts is now using them in their forecasts (Aeolus-ESA-Portal-forecast, 2020). The success of the Aeolus mission is then validating the concept of space based wind measurements using high spectral Doppler analysis on backscattered molecular signal.

Other priorities in atmospheric observation from space have been identified on the retrieval of aerosol and cloud radiative parameters as well as precipitations, which are addressing meteorological and climate studies (Stephens et al., 2002, 2018, Illingworth et al., 2015, National Academies, 2018). This need has led to new observations from space such as those proposed by the NASA-CNES Cloud-Aerosol Lidar and Infrared Pathfinder Observations (CALIPSO) mission (Winker et al., 2010) launched in 2006 and still in operation for the survey of aerosol and cloud properties and their radiative impact. It had also stimulated the development of the High Spectral Resolution Lidar (HSRL) technique for atmospheric studies from ground and from aircraft, as HSRL offers the capacity of retrieving the particle extinction and backscatter coefficients without using a priori assumptions. After the first pioneering developments using Fabry-Perot interferometers (Shipley et al, 1983; Sroga et al, 1983), the first operational airborne HSRL systems were developed in the USA at U. Wisconsin [Eloranta et al., 2008] and at NASA [Hair et al., 2008], as well as in Europe at DLR [Esselborn et al., 2008], all these systems being based on the Iodine cell absorption technique. The new Nasa/LaRC HSRL-2 system now also operate at 355 nm using a Michelson interferometric technique that was used for multispectral aerosol characterization (Müller et al., 2014). Our first airborne backscatter lidar was developed in the 90's in the frame of the French CNES-CNRS project LEANDRE (Lidar pour l'Etude des Aérosols des Nuages et du RayonnEment) for atmospheric studies. Since it was operating at similar wavelengths, it was involved in the validation of the Cloud-Aerosol Lidar with Orthogonal Polarization (CALIOP)-CALIPSO observations of cirrus clouds (Mioche et al., 2010). It was then upgraded to add a HSR channel and renamed as LEANDRE-New Generation (LNG). In contrast to the DLR and NASA systems, the French multi-wavelength system LNG was designed to operate as a HSRL at 355 nm for aerosol/cloud profiling with an added Doppler capacity for wind measurements. The HSR Doppler (HSRD) Lidar design is based on the use of a single Mach-Zehnder Interferometer (MZI) with four detection channels in phase quadrature (Bruneau and Pelon, 2003, hereafter BP03). It was chosen to better meet objectives of combined radar-lidar atmospheric observations (Delanoë et al., 2012). HSRD-LNG observations have been successfully validated and involved in several field experiments including the Aeolus CAL/VAL and the Earth, Clouds and Aerosols Radiation Explorer (EarthCARE) preparation (Bruneau et al., 2015, Schäfler et al, 2018, Cazenave et al., 2019). In the United States, Ball Aerospace developed a Doppler wind lidar including a spectral discriminator based on the same principle of Quad-channel MZI (Grund et al., 2009) which was successfully operated onboard the NASA WB57 aircraft (Tucker, 2018) and further involved in the Aeolus CA/VAL (Tucker et al., 2019, 2020).

The successes of the AEOLUS and of the CALIPSO missions have proved the potential of lidar sounding from space and its importance for meteorological and climatological applications. Many technical challenges for UV operation within AEOLUS have now been overcome by Airbus Defense and Space and ESA that pave the way for the EarthCARE mission focused on clouds and aerosols. This mission, developed in collaboration with the Japanese space agency, is to be launched by ESA in 2022. The EarthCARE payload, embarking a UV HSRL (ATLID) and a new cloud radar, should extend and improve upon

CALIPSO backscatter lidar and CloudSat radar instruments. These missions are designed for radiation budget analysis through the retrieval of cloud and aerosol properties (Illingworth et al., 2015, Stephens et al., 2018).

Taking advantage of observed performance and measurement capabilities from airborne measurements, based on the HSRD-LNG system, we study in this paper a MZI-based system for a future operational space mission. The proposed design is aiming at a performance-improved operation of a UV spaceborne lidar addressing atmospheric dynamics as a first goal. The selected MZI design may also contribute to a continued radiation budget analysis after the CALIPSO-CloudSat and EarthCARE missions. The paper is organized as follows: in section 2 we recall the choices that are to be made to meet wind profiling objectives, and in the following one, we develop the lidar design and present its main characteristics, discussing its advantages. The performance assessment of the MZI-based system is then presented and discussed in a comparative way in section 4, whereas section 5 presents final considerations in a short discussion.

## 2.      Preliminary choices

As previously introduced, the requirement of acquiring high accuracy wind profiling over a broad vertical atmospheric domain is ruled by the selection of efficient Doppler analysis on the molecular backscattered signal. Only molecular scattering can provide a high signal-to-noise ratio (SNR) in lidar detection in the upper troposphere and stratosphere at the global scale. Such a capacity has proven to be of high importance to Numerical Weather Prediction (Baker et al., 2014, LePichon et al., 2015, Rennie and Isaksen, 2020). As intensity of the Rayleigh scattering cross-section is inversely proportional to the fourth power of the emission wavelength, the use of a UV system is preferred to optimize lidar profiling.

The ADM-Aeolus main requirements are based on this approach to perform accurate wind measurements in clear air over the whole troposphere (ESA publication SP-1233 (4), 1999, ESA Aeolus MRD, 2016). The Aeolus instrument is based on newly developed narrow line UV solid state laser source, and takes advantage of the direct detection of high molecular scattering signals at 355 nm through a large diameter telescope and a cascade spectral interferometric discriminator complemented by high efficiency detectors (Aeolus-ESA-Portal-mission, 2002, ESA SP-1311, 2008). The use of high performance Accumulation Charge Coupled Devices (A-CCD) in ALADIN offers a very high efficiency (above 75 %) and a low read noise which allow near quantum noise limited detection in the UV. The ALADIN design is thus meeting most of the needs for wind measurement, but operation has revealed performance constraints set by the interferometer choice.

The first wind measurements by lidar on molecular scattering have been performed at the Observatoire de Haute Provence (OHP) in the upper-atmosphere with a double-edge (DE) technique using a Fabry-Perot (FP) interferometer (Chanin et al., 1989, Garnier et al., 1992). This photometric differential technique uses two spectral channels precisely positioned on each side of the molecular spectrum. Temperature and pressure are significantly varying on the vertical in the atmosphere and the dependence of the molecular spectral broadening with temperature and pressure needs to be accounted for in the analysis process (Dabas et al., 2008, Zhai et al., 2020). Particulate backscattering can furthermore be contributing to within a few percent and even more in the troposphere. This has been shown to be a constraint for DE-FP techniques, requiring independent

backscatter ratio measurements for bias correction (Souprayen et al, 1999a, 1999b, Witschas et al., 2020). In addition to the DE-FP of the Rayleigh channel dedicated to the analysis of the molecular return, ALADIN uses a Fizeau interferometer for the Mie channel dedicated to the analysis of particulate backcasttering. One important question in the optical design of the interferometers is to match the aperture of the telescope without degrading overall performance in case of misalignment. Both DE-FP and Fizeau interferometers have a small angular acceptance. The optical adaptation of the interferometers to the large telescope aperture is resulting in a very small field-of-view and imposes a high accuracy requirement on the transmitter-receiver co-alignment. The solution used in ALADIN is to take advantage of the large telescope magnification for both the emission and the reception to reduce alignment sensitivity. The emission beam is sent through the telescope using a polarization by-pass for separating emission and reception paths (Aeolus-ESA-Portal-mission, 2002). A drawback of this optical design is that only the backscattered light co-polarized with the emission can be detected. This is leading to significant losses in the detection of particulate backscattered signal in the case of dust and ice crystals that are inducing large depolarization. It also introduces an important limitation in the retrieval of the aerosol and cloud parameters using the analysis of the Fizeau interferometer A-CCD channels (Flamant et al., 2008).

Two-wave interferometers such as Michelson (MI) or Mach-Zehnder (MZ) interferometers are a second group of techniques that can be used in Doppler lidar measurements. MI and MZ interferometers in photometric or fringe imaging modes have already been proposed in HSRL systems for wind measurements or scattering analysis (Bruneau, 2001, BP03, Cézard et al., 2009, Liu et al., 2012, Bruneau et al., 2015, Cheng et al., 2015, Herbst and Vrancken, 2016, Tucker et al., 2018). These interferometers can include a field-compensation design (Bouchareine and Connes, 1963) which allows a large incident field angle and facilitates their accommodation with a large beam étendue system (i.e. a large size telescope and a relatively wide field angle, see Appendix B for further discussion). MI and MZ interferometers can be designed to operate with two or four detection channels (Bruneau, 2001). Dual channel techniques are very similar to the DE-FP and present the same limitations due to their sensitivity to temperature, pressure and aerosol scattering. The quadri-channel detection technique (BP03) which provides four signals in phase quadrature, allows independent determinations of interference modulation and phase, with the wind measurements being derived from the interference phase. Both particulate and molecular returns produce the same interference phase and therefore the same wind measurement. Particulate backscattering thus improves the SNR and the wind speed accuracy without introducing bias. It can also appear convenient to use the linear fringes interferometric pattern imaged on a CCD array as its position depends linearly on the Doppler shift (Herbst and Vrancken, 2016). It has been shown that fringe imaging and quadri-channel techniques provide the same theoretical measurement precision (Bruneau, 2002, BP03). It must be also considered that the accommodation of linear fringes on a square image zone with a circular aperture receiver necessitates a truncation of the aperture that would decrease the signal and hence the measurement precision. We therefore rule out the fringe imaging technique from our choices.

In addition, the quadri-channel techniques offer another advantage, as they do not require any specific spectral positioning of the laser emission relative to the interferometer, provided that a reference signal characterizing the frequency at the emission is acquired. Appropriate frequency stability is just required during the signal accumulation needed for a single measurement.

A large spectral drift of the laser source and the interferometer is tolerated over a longer timescale. Off-axis MI interferometer (Liu et al., 2012) operation associated with quadri-channel detection presents the same advantages in terms of measurement robustness: insensitivity on molecular spectral shape and particulate backscattering, tolerance with regard to spectral drifts and

misalignments (thanks to a field compensated design). Although the MZ optical arrangement is slightly less simple than the off-axis MI's, the latter brings the difficulty that one output port is necessarily very close to the input one and would hamper the positioning of the detectors. The QMZ technique thus appears as a near-optimal choice in this group, and we made this choice for optimizing the new receiver design.

An experimental comparison has been performed between the Dual-channel MZ (DMZ) and the DE-FP techniques at the OHP

(Bruneau et al., 2004). It was shown that, in the same conditions of measurement, the DMZ statistical error is lower than that of the DE-FP by a factor of 1.4 thanks to a higher optical efficiency (factor 2). As the Quadri-detection MZ (QMZ) produces a statistical error larger than that of the DMZ by the same factor (Bruneau, 2001), the QMZ and the DE-FP are equivalent in terms of statistical error. It is to be further noted that an optimized and original optical design is used in ALADIN: the reflected signal from one edge of the DE-FP interferometer is reinjected into the other edge. This scheme improves the DE-FP optical

efficiency as compared to the common method that splits the incident signal on two halves of the FP aperture, as done in the OHP setting. It leads to a theoretical advantage for the ALADIN concept in terms of measurement precision, as compared to the OHP DE-FP (and consequently to the Quadri-detection technique), by a factor that depends on the efficiency of the double-pass optical setting but cannot be larger than 1.4 (limit case of a factor 2 gain in the transmitted energy).

**Table 1: Summary of advantages (+) and disadvantages (-) of measurement techniques for direct detection wind profiling on molecular return.**

| Doppler measurement technique | DE-FP ALADIN design | QMZ Proposed design |
|---|---|---|
| Sensitivity to molecular spectral shape | Yes (-) | No (+) |
| Sensitivity to particulate scattering | Yes (-) | No (+) |
| Sensitivity to lidar alignment | High (-) | Small (+) |
| Spectral stability requirement | High (-) | Low (+) |
| Statistical error for an optimized design | $\sigma_0$ (+) | $\sigma_0 \leq \sigma < 1.4\,\sigma_0$ (-) |

Comparison of advantages and disadvantages guiding the choice of interferometric design for wind measurement on the molecular scattering is summarized in Table 1. In contrast with ALADIN, which utilizes a second interferometer (Mie channel)

for regions of the atmosphere where the aerosol load is significant, and taking advantage of the QMZ insensitivity to particulate backscattering (in terms of bias) we chose to use a unique interferometer to probe all the layers of the atmosphere. This choice allows a simplified receiver design considered to ensure good robustness and reliability. In the conducted analysis, we will

rely on numerical simulations and on results obtained from QMZ operation with the airborne HSR-LNG to define realistic parameter for the interferometer and derive the performance assessment.

An additional advantage of the QMZ technique is that it also allows measurements of the scattering ratio (ratio of the total to molecular backscatter coefficients derived from the contrast of the interference). Provided the emission linewidth is sufficiently narrow, the particulate signal produces an interference contrast near unity, significantly higher than that produced by the molecular signal (see appendix A, Eq. A14). The measurement of the interference contrast given by the total atmospheric signal then leads directly to the scattering ratio and from there to a precise quantification of the aerosol backscatter coefficient.

We will come back on this advantage for the analysis of aerosol and cloud properties.

## 3. Lidar design

### 3.1. General architecture and characteristics

As mentioned earlier, the goal for the new design is to simplify and ruggedize the existing one as much as possible, keeping as many solutions based on Aeolus and ATLID designs as needed, and accounting for recommendations made by the Aeolus

Science Advisory Group (ASAG, 2020). The transmitter is thus based on the nominal Nd:YAG laser in a redundant design with frequency converters to 355 nm, as developed for Aeolus. As the field-compensated MZI acceptance angle is much wider than that of the DE-FP, a very small emission divergence and a mono-static transmitter-receiver is no longer required. So, in contrast to ALADIN, but similarly to the ATLID lidar for EarthCARE (Hélière et al., 2012), a bi-static design is chosen. The schematic of the lidar optical design is presented in Fig. 1. The lasers can be coupled to a small telescope (typically about 15

cm in aperture) that assures an emitted beam divergence of 50 µrd for a 99 % relative encircled energy. The lasers and the expanders can be mounted on the same plate, as was done on CALIPSO, and boresight mirrors can be tilted to maintain alignment with the receiving optics. The emitted energy will conservatively be kept similar to the actual ALADIN laser source (e.g. 65 mJ @355nm). We however will extrapolate, with little risk, the possibility to extend the laser repetition rate to 100 Hz. A short laser pulse duration of 5-7 ns (corresponding to a high energy extraction from a Q-Switched oscillator and leading

to a high frequency conversion efficiency) is allowed to keep an appropriate spectral linewidth (less than 150 MHz). We will also assume that pulse linewidth and the spectral jitter or drift allows maintaining the average spectral width within 200 MHz during an accumulation of 50 shots as observed in spaceborne Aeolus observations (Reitebuch et al., 2019).

The receiver telescope with a 1.5 m aperture is coupled to a silica-silica optical fiber through an assembly of front optics which includes a 0.1 nm-bandwidth background filter. The signal fiber, with a core diameter of 340 µm (and a numerical aperture of

0.22) is used as the field stop and defines a reception field angle of 100 µrd which allows a good margin for the lidar emission-reception alignment. Note also that the receiver is not sensitive to the polarization state of the collected light and hence to the depolarization by particles scattering. As discussed in the literature, it could be useful to measure the particulate depolarization to better identify cloud and aerosol types as emphasized by the Aeolus SAG (ASAG, 2020) -see also for example Burton et al, 2012-. This can be handled inserting a polarization splitter between the telescope and the interferometer in the front optics

 system (see Fig. 1). The interferometric analysis is done in this case on the parallel polarized signal, and the perpendicularly polarized signal is measured in an additional channel as done in the DHRS-LNG instrument (Bruneau et al., 2015). However, we will not discuss a polarization channel addition in the basic lidar design, keeping it as an option. One could refer to Bruneau et al. (2015) for further information on this implementation.


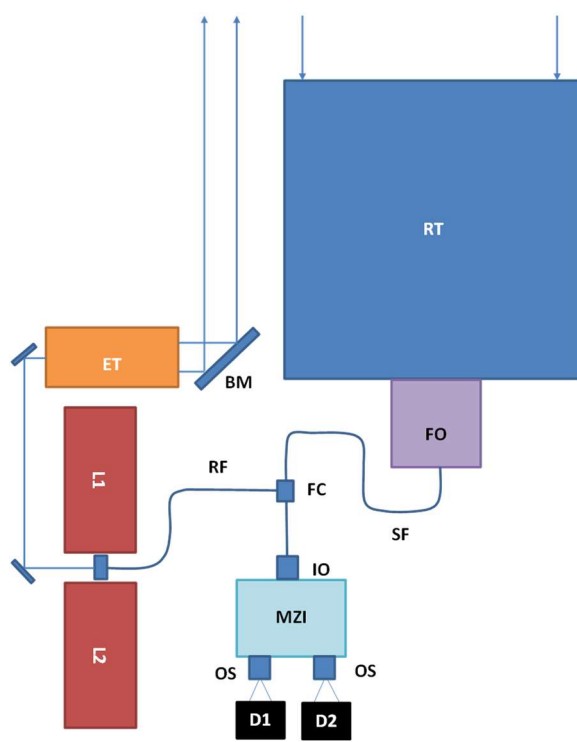

**Figure 1 Diagram of the lidar system. L1, L2 lasers (nominal and redundant), ET emission telescope, BM boresight mirror, RT receiver telescope, FO front optics (including background filter) SF signal fiber, RF reference fiber, FC fiber coupler and mode scrambler, IO input optics, MZI Mach-Zehnder interferometer, OS output separation optics,**
**D1, D2 A-CCD detectors.**

An optical mode scrambler is inserted on the fiber path in order to obtain a uniform illumination distribution at the interferometer input independently of the illumination distribution in the telescope focal plane (field stop). The mode scrambler consists in two lenses, the first lens images the output of the first fiber on the second lens while the second lens images the
first lens aperture on the second fiber. This way, the near and far fields at the output of the second fiber are uniform and fill the whole fiber beam étendue. This arrangement ensures that the interferometer response is not biased by transmitter/receiver misalignment. A small plate and a symmetrical lens arrangement inserted in the mode scrambler allows the injection of a small amount of the emitted pulse used as the reference signal and transported by a second optical fiber. The fiber and the scrambler

ensure the complete depolarization of the signal before it arrives on the MZI (even when including a polarization splitter in the front optics). This was implemented and successfully tested on the airborne LNG system. The overall efficiency of the fibers and mode scrambler on the atmospheric signal path is around 80%. The output of the fiber is then collimated by a 15-mm-focal-length lens at the MZI input port.

The main characteristics of the proposed lidar design are reported in Table 2 and compared to the nominal (Paffrath, 2006; Aeolus-ESA-portal, 2008) and actual (Reitebuch et al., 2019) ALADIN parameters (see also ASAG, 2020). This table also includes parameters discussed in next sub-sections.

**Table 2: Main characteristics of the lidar design used for the analysis of performance. Parameters for the proposed system as compared to those from the on-orbit Aeolus, as reported by Reitebuch et al., 2019.**

| System Parameters | Aeolus nominal and actual values | Our Study |
|---|---|---|
| Satellite altitude | 395-425 km<br>320 km (in space) | 400 km |
| Satellite speed | 7.2 km.s$^{-1}$ | 7.2 km.s$^{-1}$ |
| Line-of-sight angle | 35° | 45° |
| Horizontal resolution | 50 km (planned)<br>90 km (in space) | 50 km |
| Vertical resolution | 0.5/1/2 km | 0.5 km<br>(0.25 km optional in the PBL) |
| Wavelength | 355 nm | 355 nm |
| Emitted energy<br>(laser source) | 130 mJ (planned)<br>65 mJ (in space) | 65 mJ |
| Repetition rate | 100 Hz (planned)<br>50 Hz (in space) | 100 Hz |
| Allowed transmitter linewidth<br>(over 50 shots ) | 30 MHz | 200 MHz |
| Transmitter optical efficiency | 0.66 (monostatic ) | 0.9 (bistatic) |
| Telescope aperture | 1.5 m | 1.5 m |
| Telescope field of view | 0.02 mrd | 0.1 mrd |
| Receiver optical efficiency | 0.42 (polarization by-pass) | 0.5 (optical fiber) |
| Blocking filter bandwidth | 1 nm | 0.1 nm |
| Doppler sensor | 2 interferometers: Double-Edge Fabry-Perot and Fizeau | 1 interferometer: Quadri-channel Mach-Zehnder |

| A-CCD background | 6 p-e /pixel / 50 shots | 6 p-e /pixel /50 shots |

## 3.2.    QMZ Design

The field-compensated QMZ design has been detailed in several papers (Bruneau, 2001, BP03, Smith and Chu, 2016, Tucker et al., 2018). Making the choice of a unique interferometer, the optical path difference (OPD) $\Delta$ between the two arms of the interferometer is the key parameter for optimizing the wind speed retrieval over the whole troposphere using both the molecular and the particulate backscattering. In contrast with BP03 that focused mainly on the HSR separation of molecular and particulate backscattering, we give here priority to the optimization with regard to wind measurement and consider the particulate backscatter coefficient as a by-product (though of great interest). We will then consider performance obtained on the analysis of aerosol and cloud properties for this wind performance driven design.

The principles of the lidar interferometric system are presented in Fig. 2a. The optical signal incident on the interferometer is previously depolarized by passing through a multimode optical fiber. A quarter-wave plate is inserted in one arm of the interferometer and polarizers at its output separate the signal in four interference components in phase quadrature.

The interferometer is built in the same way as for the HSR-LNG (Bruneau et al., 2015) with an assembly of fused silica prisms and plates (Fig 2b). For the selected MZ optical path difference (see next section) the total prism assembly stands within a volume of approximately 1.5 cm x 2 cm x 4 cm. The different elements can be optically contacted resulting in a single block interferometer. The interferometer adjustment is obtained by construction and is unalterable.

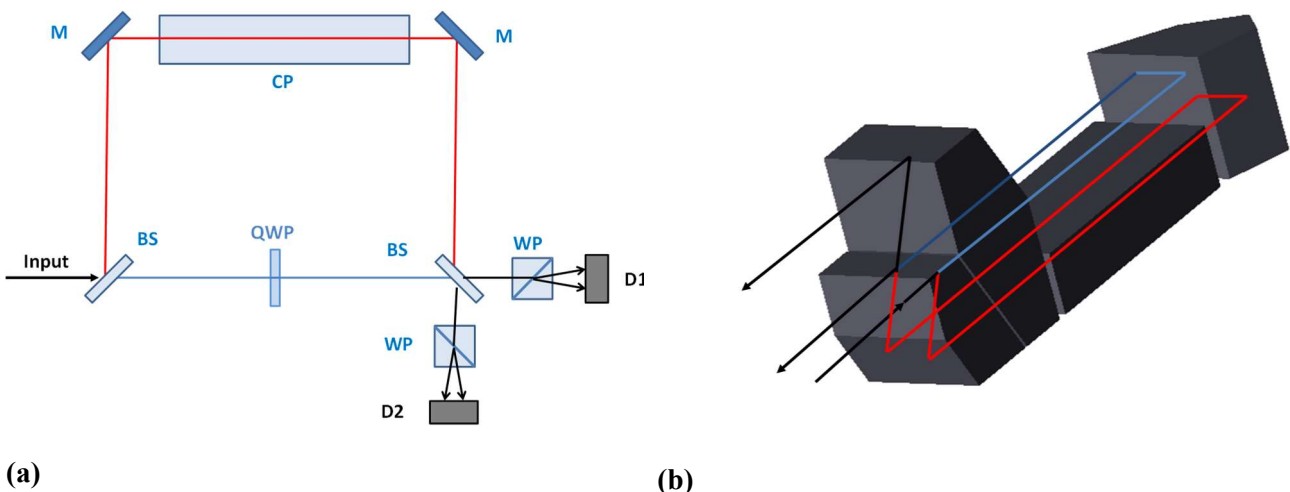

**(a)**                                        **(b)**

**Figure 2 (a) QMZ schematics: BS beam splitter, M mirror, CP field compensation plate, QWP quarter-wave plate, WP Wollaston polarizer, D1, D2 A-CCD detectors; (b) MZ actual prism arrangement (QWP, WP and detectors not shown here).**

An alternative design of a field-compensated MZI is given by the cat-eye arrangement (Tucker et al., 2018). This fully reflective design eliminates the light path through the glass which can cause a wavefront distortion but leads to a bulkier arrangement which requires a high mechanical stability. On the opposite the all-prism MZI design is insensitive to vibrations but is sensitive to temperature gradients. Nevertheless, as discussed in section 3.4, for a small OPD the all-prism design can achieve high quality and the thermally induced wavefront distortion can be easily controlled. As mentioned above, the MZI can be replaced by an off-axis Michelson interferometer but still in quadri-channel detection. In this case, the quarter-wave plate should be replaced by a 8th-wave plate used in double path. One difficulty of this design is the accommodation of one detector close to the input port of the interferometer (the off-axis angle must be small in order to achieve a field compensation without residual wavefront distortion). An additional fiber between interferometer and detector could be a solution. The alternative choice of a quadri-channel off-axis Michelson interferometer would lead to the same equations and results. We however keep considering in this paper the QMZ as the nominal design, as we refer to proven optical solutions from our airborne system.

The beam issued from each of the two MZI output ports is focused on an A-CCD detector by means of a 6.5-mm-focal-length lens. A quartz Wollaston polarizer, positioned just ahead of the focusing lens, separates the two polarizations issued from one MZI output port with an angular separation angle of 25 mrd. This way the fiber output is imaged on the detector as two spots of 150 µm in diameter separated by 160 µm. Thanks to the mode scrambler these images are well defined disks of homogeneous illuminations. Each spot fills an 8 by 8 pixel area in the A-CCD image zone in the same way as the Rayleigh channel of ALADIN. The two A-CCD provide the four channel signals (Eq. 1) of the QMZ technique.

The average signal delivered by each channel of the A-CCD outputs, for i = 1 to 4, can be written

$$S_i = \frac{S_{tot}}{4} a_i \left[ 1 + M_i M_{atm} sin \left( \varphi + (i - 1)\frac{\pi}{2} \right) \right] + Sb_i \qquad (1)$$

with $S_{tot}$ the total signal, and $a_i$, $M_i$ the relative photometric sensitivity and instrumental interference modulation of channel $i$, respectively. $M_{atm}$ is the interference modulation given by the atmospheric backscattered signal and $\varphi$ the interference phase. $Sb_i$ is the background signal due to solar detected light and A-CCD intrinsic noise.

The interference phase, from which the wind speed is derived, and the interference modulation, which leads to the scattering ratio, can be determined independently from these four signals. Actually, the line-of-sight (LOS) wind speed is calculated from the phase difference between the atmospheric signal and a reference signal obtained on a highly attenuated pick-up of the emission.

The equations which give the wind and backscatter products, as well as the respective random errors are detailed in Appendix A. These equations are the basis of numerical simulations that have been performed to analyze measurement performance of the selected design. The statistical error on wind is depending on the OPD and interference modulation (see Appendix A). We will first examine the optimal determination of these two parameters more directly linked to the interferometer design and then discuss (in Section 4) measurement SNR depending on the lidar system and mission parameters, as derived from the numerical simulations performed.

### 3.3    QMZ optical path difference

As seen from Eq. A16 in Appendix A, the statistical error on wind measurement for a given SNR is inversely proportional to the OPD and to the interference modulation $M_{atm}$. The interference modulation $M_{atm}$ is in counterpart a fast decreasing function of the OPD for a given signal spectral width. A compromise is thus to be achieved between the frequency discrimination (large OPD) and the interference modulation (small OPD). It has been shown (Bruneau, 2001) that, for measurements in a pure molecular atmosphere at $\lambda = 355$ nm, the optimum OPD is 3.2 cm. As we want to optimize measurement in conditions of clean air (scattering ratio smaller than 1.01), the OPD should be close to this value. The optimal OPD is thus quite smaller than the one proposed in BP03 (10 cm) that aimed at a compromise between wind and scattering ratio measurements. A complete parametric study is presented in Sect. 4. Such a small-OPD MZI, can be easily integrated as a solid state device derived from the design used for the airborne system (Bruneau et al., 2015), as reported in Fig. 2b.

For a 3.2 cm OPD, the modulation produced by the molecular return $M_{mol}$ is dominated by the Doppler spectral width of the thermal motion of molecules and varies from 0.49 to 0.59 depending on the atmospheric temperature with an average value of 0.56. Note that at high atmospheric pressure, near the surface, Brillouin molecular scattering causes a relative broadening of this spectrum by approximately 1.7 %. This effect causes itself a small decrease in the molecular modulation $M_{mol}$ from 0.49 to 0.48. As the spectral broadening caused by the particle Brownian motion is small compared to the emitted linewidth, the modulation produced by the particle return $M_{par}$ is very close to that given by the emitted linewidth. For an emission linewidth of 200 MHz, including the broadening by frequency shift or jitter during the measurement integration duration, and a 3.2 cm OPD, $M_{par}$ is still as high as 0.95, according to equations given in BP03.

Note again that, as the QMZ technique allows the determination of wind speed (through the interference phase) and the scattering ratio (though the interference modulation) independently, the superposition of a narrow particle spectrum to the wide molecular spectrum does not bring any measurement bias. The increase in interference modulation caused by an additional particle scattering will indeed improve the wind measurement accuracy.

### 3.4    QMZ optical quality

The interferometer optical quality is obviously an important aspect of the design as it determines the intrinsic modulation factors $M_i$ (Eq. 1). These modulation factors can be slightly different from one channel to the other but for clarity we consider here that they are all equal to $M_0$. The equations of Appendix A show that the random errors (both on wind speed and scattering ratio) are, at first order, inversely proportional to this intrinsic modulation factor $M_0$.

We can also consider that $M_0$ is the product of two factors: $M_W$ the modulation degradation caused by imperfections of optical surfacing and non-homogeneities of the optical index resulting in a global wave-front error (WFE) and $M_R$ the modulation degradation caused by the imbalance of the reflection-transmission factors of the beam-splitters.

It has been shown that the modulation degradation of a Michelson interferometer is a function of the RMS value of the WFE ($\sigma_{WFE}$), almost independently of the shape of this distortion (Liu et al., 2012). This result is obviously also valid for a MZI. Assuming a Gaussian distribution of the WFE we obtain the modulation factor $M_W$ as

335

$$M_W = \frac{1}{\sigma_{WFE}\sqrt{2\pi}} \int exp(-\frac{\delta^2}{2\sigma_{WFE}^2})cos\left(\frac{2\pi\delta}{\lambda}\right)d\delta \qquad (2)$$

We can also admit that $\sigma_{WFE}$ is proportional to the area of the useful interferometer aperture. It is therefore essential to evaluate the useful aperture of the MZI which can withstand the large beam étendue of the system. The calculation, detailed in Appendix B, leads to the chosen lidar parameters to a useful aperture $D_{MZI}$ = 6.2 mm.

Using Eq. (2) we calculate $M_W$ as a function of the MZI aperture for different values of the $\sigma_{WFE}$ per square cm and an operation
wavelength of 355 nm (Fig. 3a). For comparison, the intrinsic instrumental contrast of the HSRD-LNG interferometer is 0.65 for a useful aperture of 20 mm (and a 20 cm OPD) that corresponds to a $\sigma_{WFE}$ less than 20 nm/cm$^2$. For the same optical quality $\sigma_{WFE}$ = 20 nm/cm$^2$ (which can be nonetheless improved) a modulation factor $M_W$ > 0.99 is achievable for the proposed MZI with a useful aperture of 6.2 mm (see Fig. 3a).

In the case of a non-homogeneous temperature distribution in the optical elements, and more particularly a transverse
temperature gradient in the compensation plate, leading to an angular tilt of one wavefront plane, we computed that the impact is an additional degradation of about 0.5 % in $M_w$ for a thermal gradient of 0.1 K per cm. This shows a relative acceptance of temperature changes, not requiring a high standard in temperature control of the interferometer.

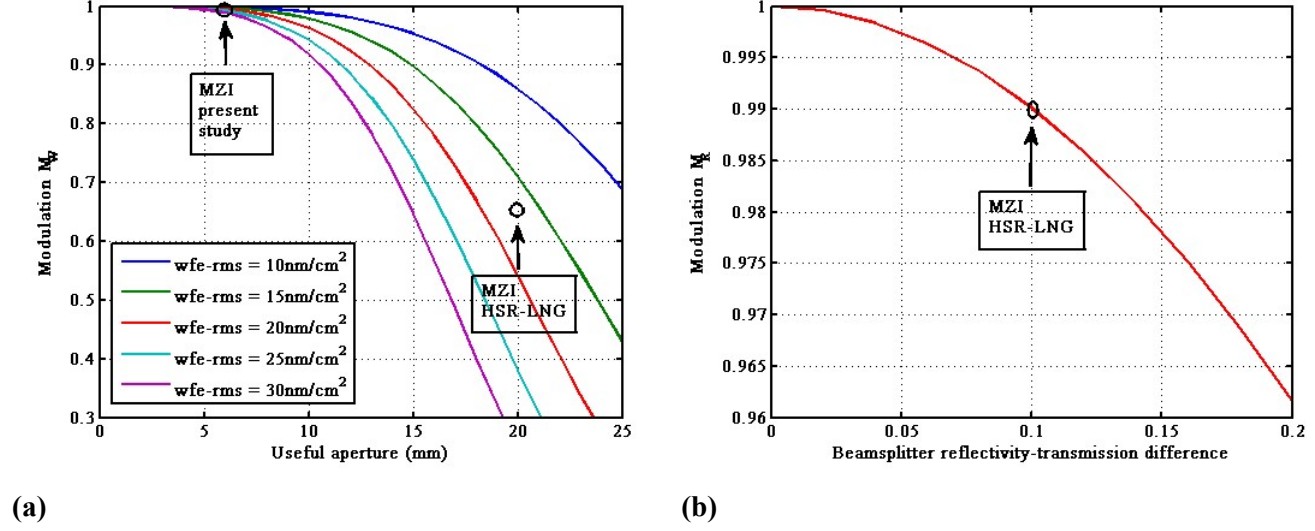

(a)                                                                (b)

**Figure 3 (a) Instrumental modulation factor $M_W$ as a function of MZI useful aperture for different RMS-WFE. The dots represent the MZI of the present study and the actual MZI of HSRD-LNG. (b) Instrumental modulation factor**
**$M_R$ as a function of the beam splitter R-T difference.**

Note also that, for our on-axis, field compensated design, the variation of OPD is only dependent on the fourth power of the source angle (the second power dependence is canceled and the third order dependence is null due to all on-axis optical elements). With the system parameters presented above, the source full angle brings a negligible contribution of less than 0.1 nm to the RMS-WFE (see Appendix B).

The second intrinsic modulation factor $M_R$ stems from the fact that it is not possible for a pure (non-absorbing) dielectric coating to ensure a perfect balance between the reflectivity $R$ and the transmission $T$ for both polarizations, at an incidence angle of several tens of degrees. The consequent imbalance of the beam-splitter reflectivity causes a modulation factor $M_{Rmin} = \frac{2RT}{(R^2+T^2)}$ on two of the four channels of the QMZ, the two others channels being unaffected. The average reflectivity modulation factor is then $M_R = \frac{1}{2(R^2+T^2)}$. Fig. 3b presents the decrease of instrument intrinsic modulation $M_R$ as a function of the reflectivity-transmission difference $R$-$T$. For comparison, in the HSRD-LNG interferometer the incidence angle on the beam-splitters is 40° and a reflection-transmission imbalance of 0.1 has been achieved leading to $M_R = 0.99$.

We can then conclude that an intrinsic instrument modulation $M_0 = 0.98$ is achievable for a MZI of small OPD (3.2 cm) associated to a space-borne lidar with a receiver aperture – field-angle product as large as 0.15 m.mrd.

Note also that the optical quality requirements on the MZI are less stringent than for a FPI, which necessitates a larger aperture, because of low field angle acceptance, and a high surfacing quality to avoid losses in transmission and finesse.

### 3.5 QMZ calibration

Prior to processing actual atmospheric signals, it is necessary to perform a calibration of the MZI for determining the transmission and modulation factors $a_i$ and $M_i$ of the MZI defined in Eq. (1). The sensitivity parameters $a_i$ can be determined by recording the four QMZ signals in the absence of interference either by masking sequentially one arm of the interferometer and the other, and adding the signals of the two sequences, or by using the laser in multimode operation (without injection seeding) to get a flat interferometric response. The calibration of the instrumental modulations $M_i$ is then obtained by recording a long sequence of reference signals with a slowly varying OPD phase (with temperature for instance) or by tuning the laser seeder frequency. The recorded signals are fitted by a least-square method on the four modeled signals (Eq. A2) by adjusting the $M_i$ parameters.

An error in the determination of the instrumental sensitivities $a_i$ and instrumental contrasts $M_i$, propagates to the calculation of the complex signal $Q$ (Eq. A4) and then to the LOS velocity and scattering ratio measurements. These calibration errors apply equally on the reference complex signal $Qr$ but with a slightly different impact due to a different modulation factor and to the Doppler phase shift. The consecutive measurement errors are interference-phase dependent with a negligible averaged value but with a perceptible amplitude.

We will discuss calibration from results obtained from airborne HSRD-LNG observations during previous field experiments in section 4.3. The impact of calibration errors on the simulation of new spaceborne measurements is further discussed in section 4.6.

## 4. Performance assessment

### 4.1 Model description

Performance modeling starts by computing the total signal $S_{tot}$ (Eq.(1)) expressed in number of photo-electrons (p-e) for a single shot, is based on the canonical lidar equation:

$$S_{tot}(R) = \frac{\eta E A T_{inst}}{h\nu} \int_{r=R-\delta R/}^{r=R+\delta R/2} \frac{T_{atm}^2(r)\beta(r)}{r^2} dr \qquad (3)$$

where $R$ (m) is the range from the instrument, $\delta R$ (m) is the range gate, $\eta$ is the detector quantum efficiency, $E$ (J) is the emitted energy, $A$ (m$^2$) is the receiver aperture area, $T_{inst}$ is the total optical efficiency product of transmitter and receiver efficiencies, $h\nu$ is the emitted photon energy, $T_{atm}$ is the atmospheric transmission from instrument to range $r$ for a line-of-sight (LOS) nadir angle $\theta$, $\beta$ (m$^{-1}$sr$^{-1}$) is the total (molecular and particulate) backscatter coefficient.

The atmospheric backscattering and transmission are computed from the ESA reference model of atmosphere (RMA) including a measurement statistics of aerosol backscatter coefficients (Vaughan et al., 1998). A cirrus cloud between 9.5 and 10.5 km altitude with an optical thickness of 0.1 (backscatter coefficient equal to $4.10^{-6}$m$^{-1}$sr$^{-1}$) can also be introduced. The RMA comprises data of different aerosol and cloud backscatter coefficient and extinction, background radiance, and ground albedo. Extinction and backscatter coefficients corresponding to the Lower Quartile (LQ), Median (MD), Higher Quartile (HQ) and Median profiles with an additional cirrus cloud (MD+CIR) are given in Fig.4 for the 355 nm operation wavelength.

A background signal $S_{bs}$ caused by the sun illumination and the intrinsic read noise $S_{bd}$ due to the A-CCD detector are added on all the range gates so that the total background noise is written as

$$S_b = S_{bs} + S_{bd} = \eta A \Omega T_{rec} I_s(\lambda) \delta\lambda \cos(\theta_s) \frac{alb}{\pi} T_{atm}(\theta_s, z_s) T_{atm}(\theta, z_s) \frac{2\delta R}{c} + S_{bd} \qquad (4)$$

where $\Omega$ (srd) is the receiver field solid angle, $T_{rec}$ is the receiver efficiency, $I_s$ (Wm$^{-2}$) is the sun irradiance at top of atmosphere at wavelength $\lambda$, $\delta\lambda$ is the bandwidth of the blocking filter, $\theta_s$ is the sun elevation angle, $alb$ is the albedo (surface or cloud) and $z_s$ is the altitude of the scattering layer. For these computations we take $alb = 0.3$ for land surface and add $alb = .015$ for thin cirrus. Assuming a dawn-dusk sun-synchronous polar orbit, similarly to Aeolus, we take $\theta_s = 80°$. The blocking filter has a bandwidth $\delta\lambda = 0.1$ nm. An important advantage of the A-CCD detectors is their high quantum efficiency ($\eta = 0.85$) combined with a low read noise. The signal is pre-accumulated on 50 shots on the A-CCD (e.g. 0.5s corresponding to an along track horizontal sampling resolution of 3.6 km, similar to Aeolus) before being read by the analog-to-digital converter that adds a noise equivalent to a background signal $S_{bd}$ of 6 p-e per detector pixel (Paffrath, 2006).

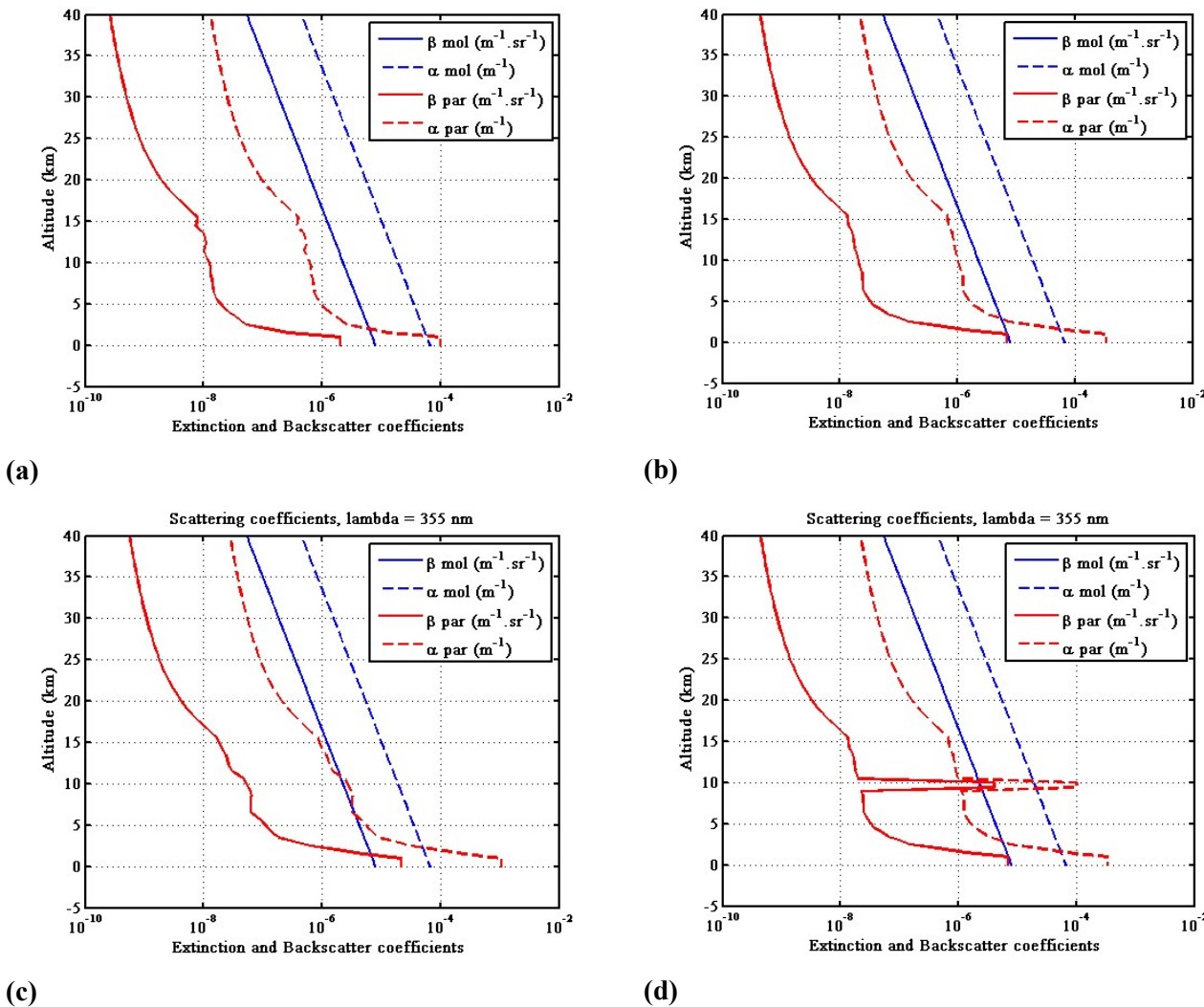

**Figure 4 Backscatter and extinction coefficients vertical profiles at 355 nm used in the performance model. (a) RMA Lower Quartile, (b) RMA Median, c: Higher Quartile, d: Median plus cirrus.**

The signals are first accumulated on the ACCD over 50 shots forming an elementary sample. Then 14 elementary samples are reference adjusted and accumulated, corresponding to a total of $Ns$ = 700 shots, for an observation horizontal resolution of 50 km and the resulting total SNR is

$$SNR = \frac{S_{tot}\sqrt{N_s}}{\sqrt{S_{tot}+S_b}} \qquad (5)$$

The recorded total signals and SNR (i.e. calculated on the sum of the four detection channels) vertical profiles used in the performance model are presented in Fig. 5 for the median (MD) RMA backscatter profile with the cirrus cloud and a 45° LOS angle. Except a large increase in the cirrus cloud and at ground, the SNR is slowly varying over the whole troposphere. The standard deviation on the projected horizontal (HLOS) wind speed is calculated from Eq. A17 as

$$\sigma(V_{HLOS}) = \frac{\sigma(V_{LOS})}{\sin(\theta)}\frac{R_E+z}{R_E+z_{sat}} = \frac{c\lambda}{4\pi\Delta s}\frac{\sqrt{2}}{(\theta)}\frac{1}{SNR}\frac{1}{M_0 M_{atm}}\frac{R_E+z}{R_E+z_{sat}}\sqrt{1-\frac{M_0^2 M_{atm}^2}{4}} \qquad (6)$$

where $R_E$ is the Earth radius and $z_{sat}$ the satellite altitude.

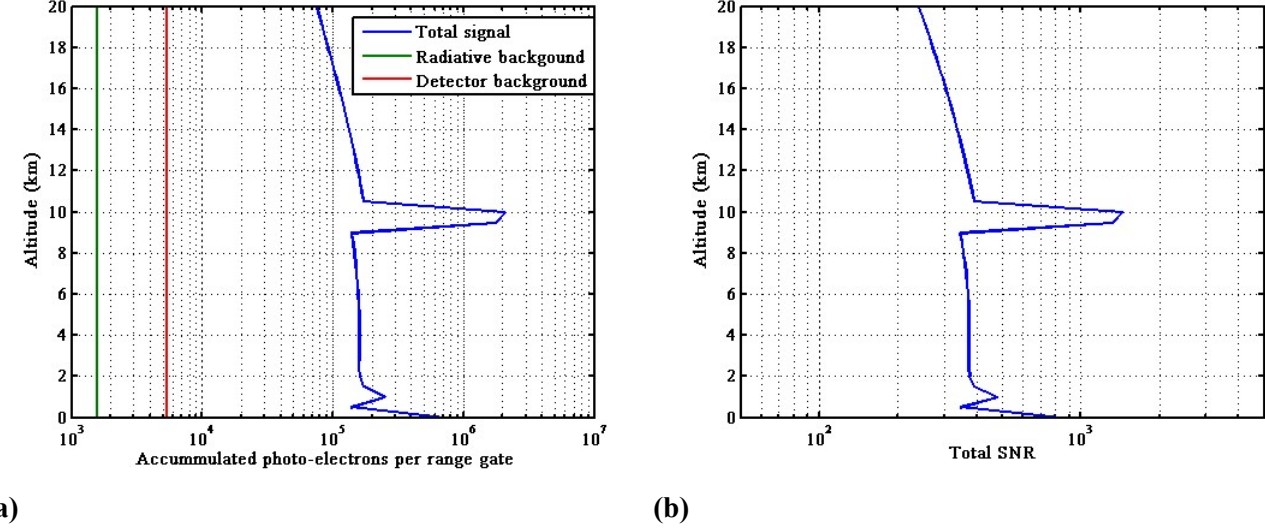

**(a)**  **(b)**

**Figure 5 (a) Number of accumulated photo-electrons per range gate (b) SNR for RMA MD backscatter profile plus cirrus at 45° LOS angle.**

The MZI intrinsic modulation is calculated assuming a $\sigma_{WFE}$ of 20 nm/cm$^2$, an $R$-$T$ imbalance of 10 % and an emitted spectral width and jitter of 200 MHz RMS. The modulation on molecular return $M_{mol}$ is computed with the standard model of temperature at mid latitudes.

The background and the reference signals are assumed to be recorded with a SNR much higher than that of the atmospheric signal, so that they add a negligible variance to the measurements when subtracted to the total signal. This can be done on each acquisition sequence to avoid possible bias due to hot pixels, as observed and corrected with A-CCDs in space (Weiler et al., 2019).

The speckle noise is not an issue for the measurement accuracy, as the design is based on a large aperture and wide field of view. The receiver number of spatial speckles $M_s$, defined as the ratio of the solid angle of the transmitted beam to the diffraction solid angle of the telescope is 2.75 10$^4$. The number of temporal speckles $M_t$ is around 500, due to the extended range gates. The total number of speckles $M_{spec}=M_s.M_t$ is then approximately 1.4 10$^7$. As the number $N$ of collected photo-

electrons per shot and range gate is on the order of $10^3$, the speckle noise (given by the ratio $N/M_{spec}$) is much smaller than 1, so that the noise variance is not modified.

Note also that the fiber beam étendue is necessarily larger than the telescope beam étendue in order to accept the received beam with a substantial alignment margin. The maximum number of spatial speckle of the fiber ($9\ 10^4$) is then larger than $M_s$ and does not limit the total number of speckle of the signal.

Appendix C shows measurements performed with HSRD-LNG where the error calculated with equations similar to those used in the performance model is compared to the experimental error. Fig. C1 shows that the calculated and experimental errors are in good agreement. A comparison between lidar wind measurements, including calculated error, and collocated radiosonde profiling is also presented in Appendix C (Fig. C2 and Table C1). A similar comparison of measured and calculated errors on the backscatter coefficient shows also a good agreement (Fig. C3). We can therefore consider that these results validate the equations used in the performance model.

## 4.2    OPD and LOS angle optimizations

We first come back on the OPD optimization to detail results previously given on average. The standard deviation of the statistical error on the retrieved horizontal wind speed $V_{HLOS}$ as given by Eq. (6) is presented in Fig. 6 for the median, higher quartile and lower quartile of the aerosol backscatter coefficient distributions and for OPDs varied from 2.2 to 4.2 cm. The LOS angle is set to 45° in these computations. Fig. 6d thus shows that the optimum OPD of 3.2 cm is effective for all atmospheric models corresponding to scattering ratios smaller than 2 as it is the case outside the boundary layer. We now look at the pointing angle. The standard deviation of the statistical error on horizontal wind speed is presented in Fig. 7 for the median, higher quartile and lower quartile of the aerosol backscatter distributions and for LOS angle varied from 35° to 55°. Results are analyzed (as for OPD) looking to HLOS wind error averaged between 0.5 and 15 km, and on vertical profiles. The MZI OPD is set to 3.2 cm on these computations.

One can see from Fig 7d that the optimal 3.2 cm OPD and 40°- 47° LOS angles are well matched to aerosols distributions from lower to higher quartiles of RMA aerosol distribution. The errors are noticeably degraded in the troposphere below 10km in altitude for angles smaller than 40° or larger than 50°. For all the aerosol distributions, a LOS angle close to 45° appears to give better results. We thus chose 45° as the optimal value for conducting final analysis performance on HLOS wind retrievals. The standard deviation of the horizontal wind speed statistical error averaged between 0.5 and 15 km in altitude is slightly less than 2 ms$^{-1}$ in the troposphere for the three distributions.

## 4.3    Comparison with Aeolus requirements

Fig 7a shows the HLOS wind speed error profiles for the selected parameters and the aerosols distributions LQ, MD, HQ and MD + cirrus for the parameters listed in Table 2, corresponding to the previously discussed choices and optimization. Fig. 8b shows the same results but for a variable vertical resolution according to Aeolus requirements. The modelled statistical errors for the RMA-MD aerosol distribution presented in the previous section are summarized in Table 4 in comparison with Aeolus

requirements. One can see that in the free troposphere the Aeolus requirements are fulfilled with a good safety margin. For a constant vertical resolution of 500 m, results obtained in the atmospheric boundary layer are better than in the clean atmosphere, due to a larger scattering ratio which compensates for the decrease in transmission, till the aerosol attenuation becomes too high. The errors are nevertheless slightly higher than the requirement (1 ms$^{-1}$) in the PBL with values around 1.5 ms$^{-1}$ on

average.

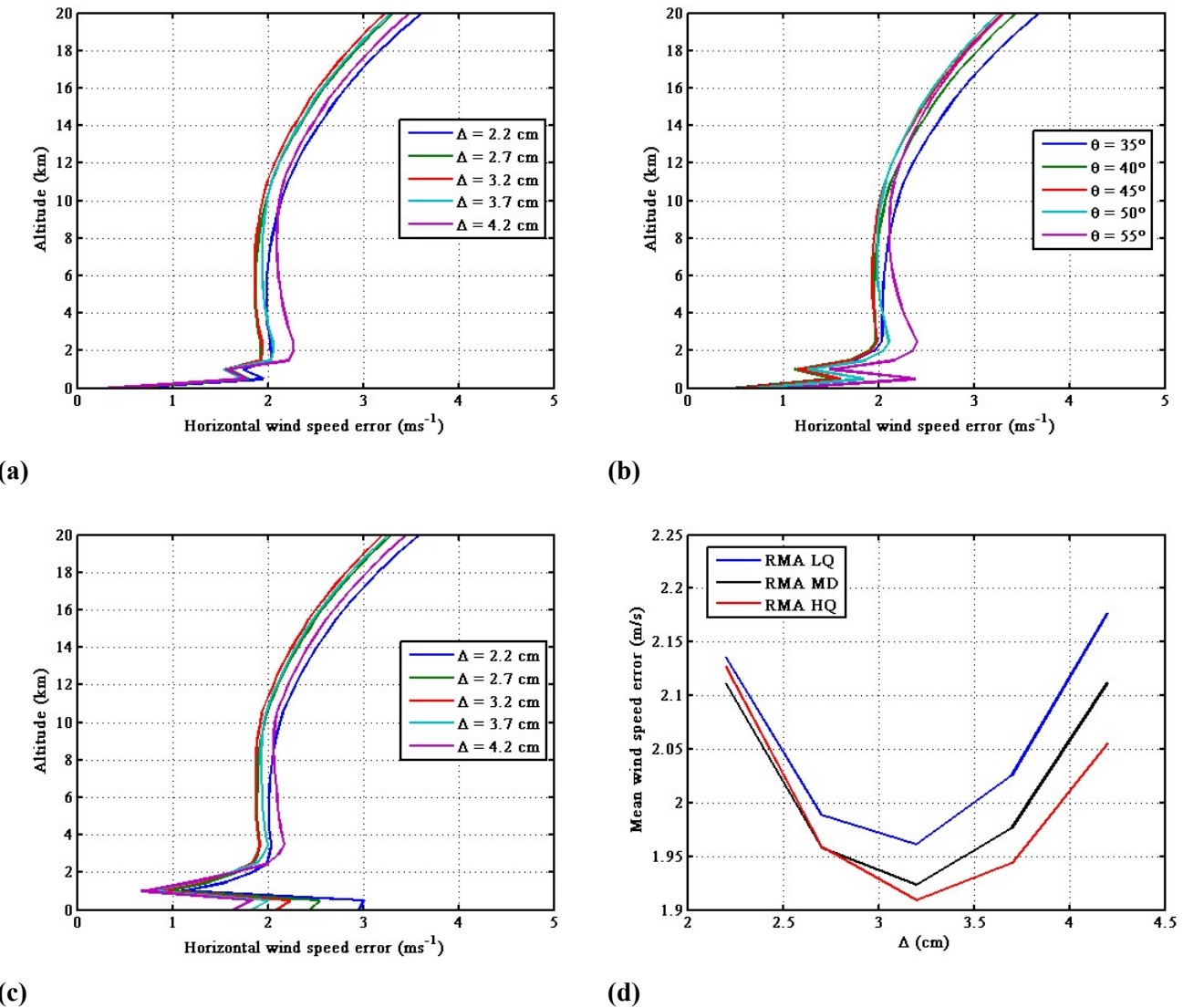

(a)                                                              (b)

(c)                                                              (d)

**Figure 6 (a, b, c) Horizontal wind speed (HLOS) statistical error as a function of altitude for OPD for Lower Quartile, Median, Higher Quartile RMA aerosols distribution respectively. (d) wind speed error averaged between 0.5 and 15 km in altitude as a function of OPD.**

In presence of a cirrus cloud, the wind speed accuracy is improved to a fraction of ms⁻¹ due to the increased signal return and interference modulation $M_{atm}$ (Eq. A3 and A12), but is degraded in clear air below the cloud layer by 0.5 ms⁻¹ due to the decreased atmospheric transmission (by about 20 % on the 2-way transmission). The effect of a cirrus cloud on wind and backscatter coefficient retrieval accuracies at different levels is discussed in Sect. 4.5 as a function of the cirrus optical thickness.

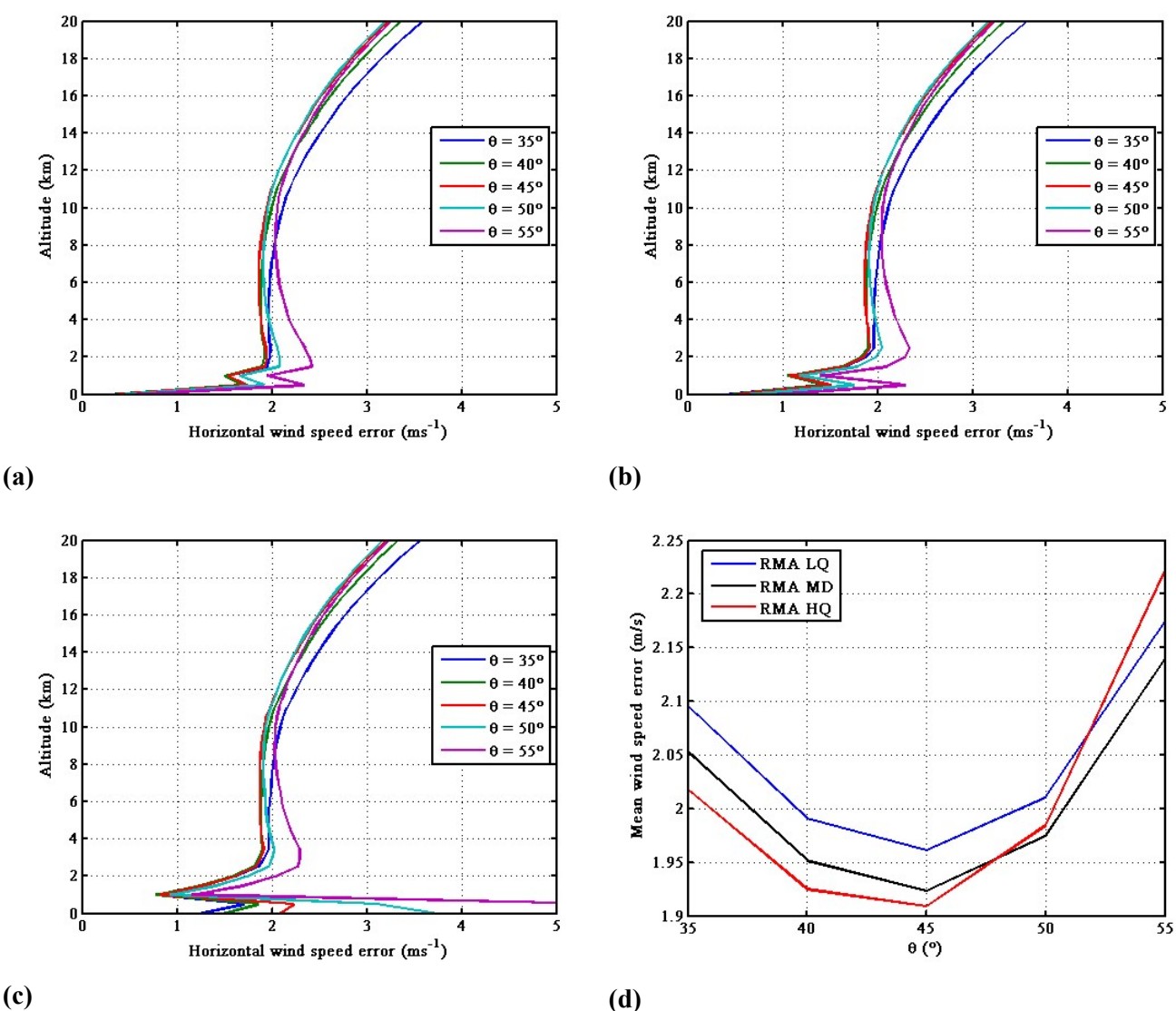

**Figure 7 (a, b, c) HLOS wind speed statistical error as a function of altitude for LOS angles between 35° and 55°, for Lower Quartile, Median, Higher Quartile RMA aerosols distribution respectively. (d) HLOS wind speed error averaged between 0.5 and 15 km in altitude as a function of LOS angle.**

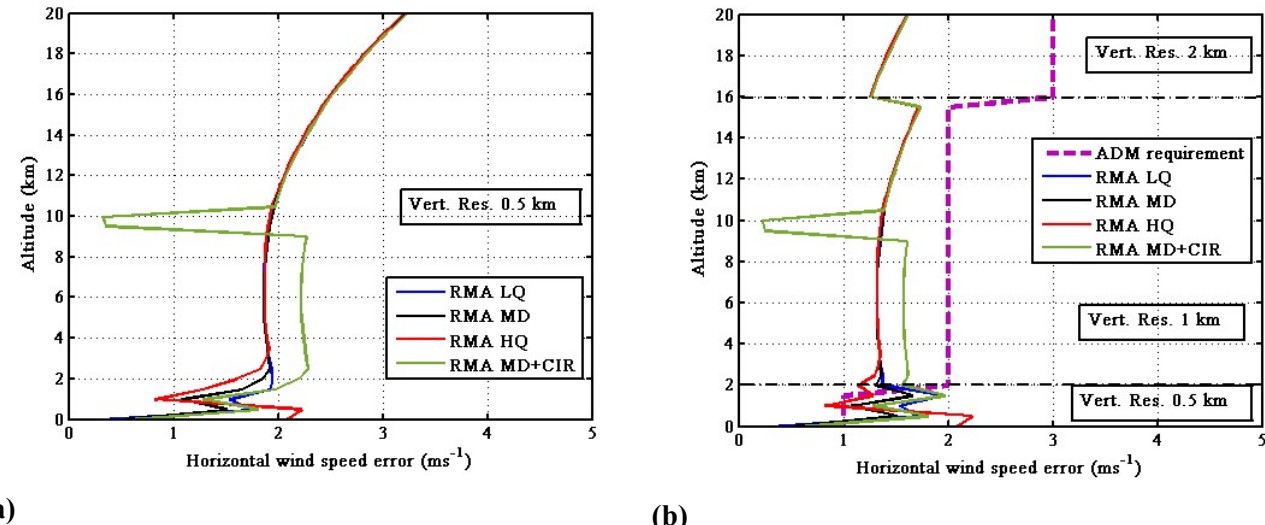

**(a)**                                                                      **(b)**

**Figure 8 Vertical profile of the horizontal wind speed error (std deviation) for different backscatter profiles: LQ, MD, HQ and MD + cirrus for the optimal parameters Δ = 3.2 cm and θ = 45°; (a) constant vertical resolution of 0.5 km; (b) variable vertical resolution (0.5 to 2 km) and requirements according to Aeolus.**

**Table 4. HLOS wind statistical error of the proposed QMZ design compared to the original Aeolus requirements.**

| Altitude (km) | Vertical resolution (m) | Aeolus horizontal wind speed accuracy requirements (ms⁻¹) | QMZ averaged horizontal wind speed statistical error modelled for RMA MD (ms⁻¹) |
|---|---|---|---|
| 0-2 | 500 | 1 | 1.5 |
| 2-16 | 1000 | 2 | 1.4 |
| 16-20 | 2000 | 3 | 1.9 |

### 4.4     Backscatter measurements

The choice of a small OPD is a strong advantage for wind measurements in clear air but is not optimized for retrieving the particulate backscatter coefficient (the OPD should be larger as discussed before, see also BP03). The QMZ technique with the selected parameters can nevertheless retrieve backscatter coefficients and determine the lidar scattering ratio $R_\beta$ (see Appendix A) with a relatively good accuracy, as we will see now. The derivation of the particulate backscatter requires the knowledge of the molecular backscatter coefficient $\beta_{mol}$ and the calculation of the molecular and particle interference

modulations $M_{mol}$ and $M_{par}$ respectively. These calculations bring necessarily additional errors in the backscatter retrieval
process. They are summarized later in this section and detailed in Appendix D.

Let us first look to noise induced errors, assuming no other random error source. Figure 9 shows the relative standard deviation of the statistical error on $R_\beta$ and on $\beta_{par}$ (according to Eqs. A18 and A19 in Appendix A) for the LQ, MD, HQ and MD with cirrus backscatter profiles.

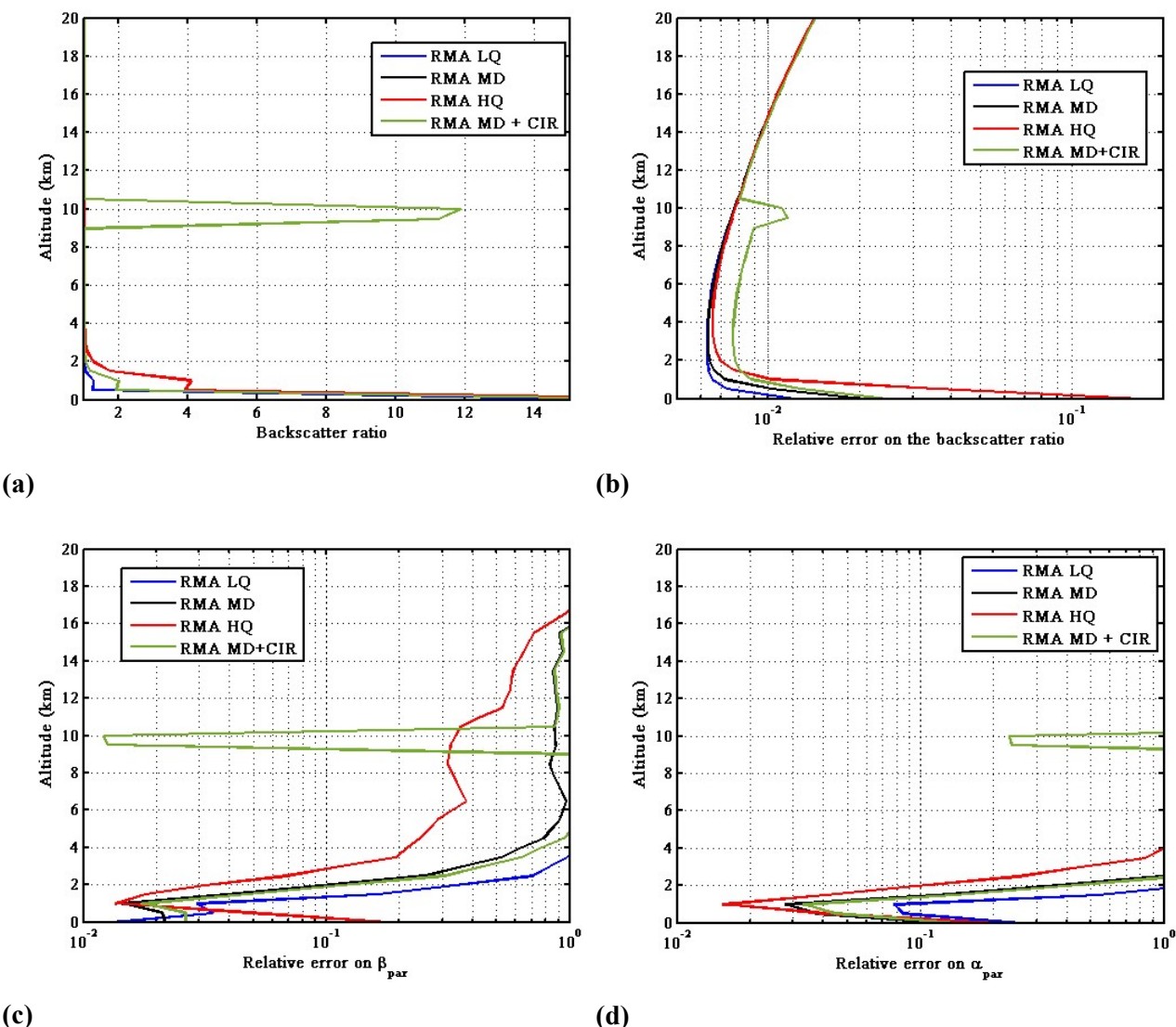

(a)                                                     (b)

(c)                                                     (d)

**Figure 9 Actual scattering ratio (a) Relative statistical error (standard deviation) on the scattering ratio measurement**
**(b) on the backscatter coefficient (c) and on the extinction coefficient (d) for LQ, MD, HQ and MD plus cirrus backscatter profiles.**

The scattering ratio $R_\beta$ can be retrieved with a relative statistical error of approximately 1 % over the troposphere except above the tropical tropopause and in the first kilometer above the surface where the error can increase up to 10 % due to the strong atmospheric attenuation for the HQ aerosol distribution. One can argue that in this case the scattering ratio corresponds to a more polluted situation ($R_\beta \sim 3$), where such a reduced precision may be more acceptable than an equivalent bias for assimilation purpose (Ma et al., 2019).

The error on $\alpha_{par}$ is somehow more important than for $\beta_{par}$ due to the simple linear signal derivative procedure used in this analysis. More suitable and more efficient methods can be applied by distinguishing, slicing and processing aerosol sub-layers one by one, taking advantage of the simultaneous retrieval of backscatter and transmission, extending further previous analyses (Young, 1995; Young et al., 2009) and possibly using variational methods. This is however beyond the scope of this paper. Nevertheless, as seen in Fig. 9d, the relative error remains smaller than 10 % in the boundary layer, using this simple derivative method.

Let us now look at the errors due to the use of meteorological re-analyses for the retrieval of the particulate backscatter. These errors, often qualified as random-systematic, are inherent to the modelling of the molecular scattering which depends on the temperature profile using an a priori guess and include random errors. They are for a large part common to all the retrieval processes whatever the measurement method. The particle backscatter and extinction retrieval (Eq. A8 –A10) requires knowledge of $\beta_{mol}$ and $M_{mol}$. The calculation of $M_{mol}$ is not as straightforward as that given by Eq. A14 since it must include the effect of Brillouin scattering. For an operating wavelength of 355 nm the Brillouin doublet separation is much smaller than the thermal linewidth. The resulting spectrum is a broadening and distortion of the Gaussian profile. A correction can be derived from a model (Tenti et al., 1974) or measurements (Witschas et al., 2010). The effect is to the first order proportional to the pressure with a maximum relative broadening of 1.7 % near ground. This broadening causes a relative decrease of $M_{mol}$ of $6.10^{-3}$ as compared to that calculated from the Gaussian thermal line. The Brillouin effect must be included in the calculation of $M_{mol}$ but the uncertainty on the actual atmospheric pressure leads to a second-order variation that can be neglected. This constraint is less critical here for the quantification of the molecular backscatter coefficient than for wind measurements using a DE-FP device (Dabas et al., 2008).

As seen in Appendix A (Eqs A14 and A15), $M_{mol}$ is sensitive to temperature and this effect may significantly contribute in clear air and must be taken into account. The molecular linewidth $\gamma_{mol}$ varies as the square root of temperature (see Eq. A15), which leads to a variation of about 20 % over the atmospheric column for a standard temperature profile. Assuming that the atmospheric temperature is known from a meteorological model to within 2K allows reducing the uncertainty on $M_{mol}$ to less than 0.5 % (see Appendix D, Fig. D2a). The other source of error on the particle backscatter coefficient comes from the modeling of the molecular backscatter coefficients which is proportional to the molecular density and then also depends on the temperature vertical profile. As a whole, the knowledge of atmospheric temperature to within 2K allows reducing the relative uncertainty on $\beta_{mol}$ to less than 1 % (see Appendix D, Fig. D2b).

Note that wind speed variation or turbulence in the probed volume could slightly reduce *Mpar*, however with a Doppler shift of 5.6 MHz per m.s$^{-1}$ the broadening of the particle backscattered spectrum is much less than the emitted laser linewidth of 100-200 MHz, and thus the impact on *Mpar* is negligible. We therefore consider that *Mpar* is not affected by the atmospheric conditions.

In clear air the uncertainty on $M_{mol}$ dominates the total error on the backscatter and extinction coefficients (see Appendix D, Eqs D1 and D2), but in regions where the aerosol load is significant ($R_\beta \geq 2$), contributions to the total error from uncertainty in $\beta_{mol}$ and $M_{mol}$ remain small and comparable in magnitude. Figure 10 shows the relative errors on $\beta_{par}$ and $\alpha_{par}$ given by a random temperature error $\sigma(T) = 2K$ at all altitudes, computed for the MD+CIR aerosol profile and a standard temperature profile. These errors must be root square summed (RSS) with the statistical errors caused by detection noise. The total relative

error on $\beta_{par}$ remains on the order of 1% in the regions where the aerosol load is significant ($R_\beta \geq 2$). The total relative error on $\alpha_{par}$ is largely dominated by the SNR random error and stays on the order of 10% in the PBL.

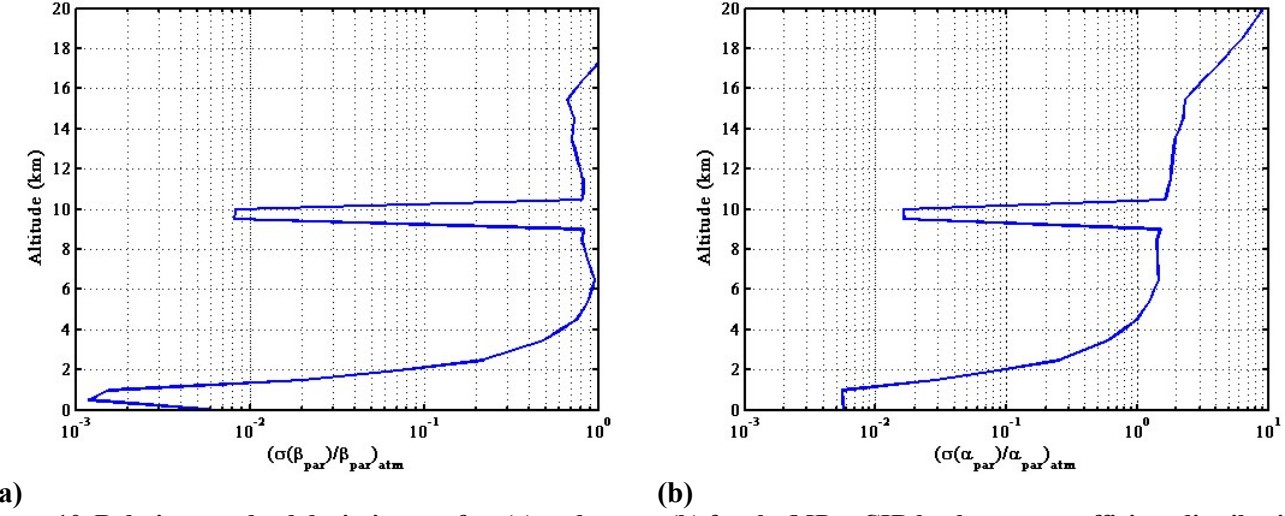

(a)                                                          (b)

**Figure 10  Relative standard deviation on $\beta_{par}$, (a) and on $\alpha_{par}$ (b) for the MD + CIR backscatter coefficient distribution caused by uncertainties of 2K in the atmospheric analysis as a function of altitude.**


### 4.5    Measurements in presence of semi-transparent clouds

The evolution of the measurement performance is modelled for the RMA-MD scattering profile with the addition of a semi-transparent (cirrus) cloud whose optical thickness is varied from 0 to 0.5. Fig.11 shows the SNR, errors in HLOS wind, backscatter and extinction coefficients at different levels. The measurements above the cloud are almost unaffected, despite

the increase of the background noise due to the increasing cloud albedo. Performance in the PBL below the cloud is degraded by the lower atmospheric transmission while performance in the cloud is improved due to the higher return signal.

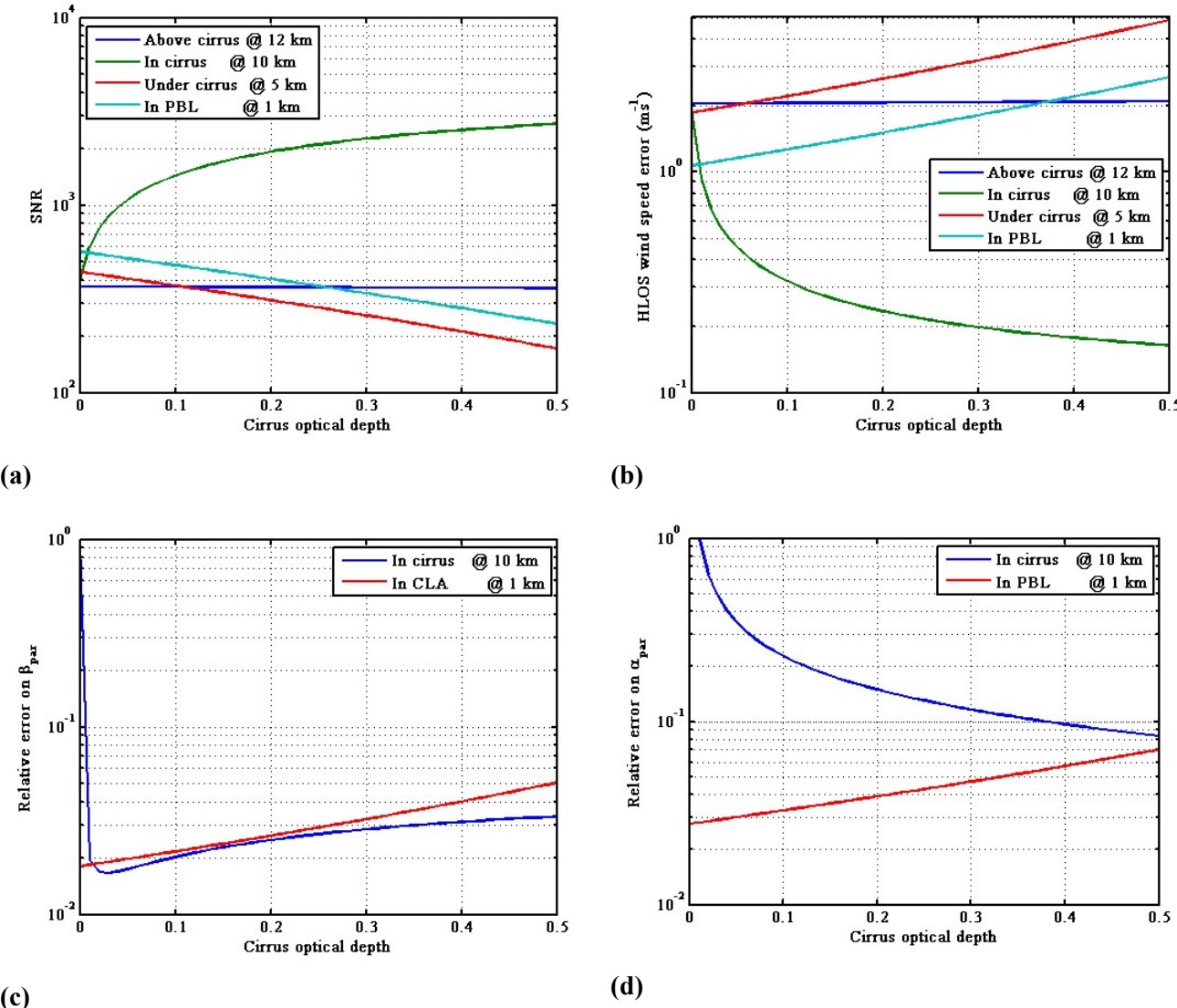

**Figure 11 (a) SNR, (b)HLOS wind speed error, (c) $\beta_{par}$ relative error and (d) $\alpha_{par}$ relative error as a function of the cirrus vertical optical depth for different measurement altitudes.**


### 4.6    Instrument calibration dependent errors

The maximal value of the LOS velocity error and the relative error on the scattering ratio measurement are presented on Fig. 12, for a $10^{-3}$ relative error in the sensitivity calibration of a single QMZ channel, as a function of the scattering ratio and for LOS wind speeds extending from 0 to 100 ms$^{-1}$.


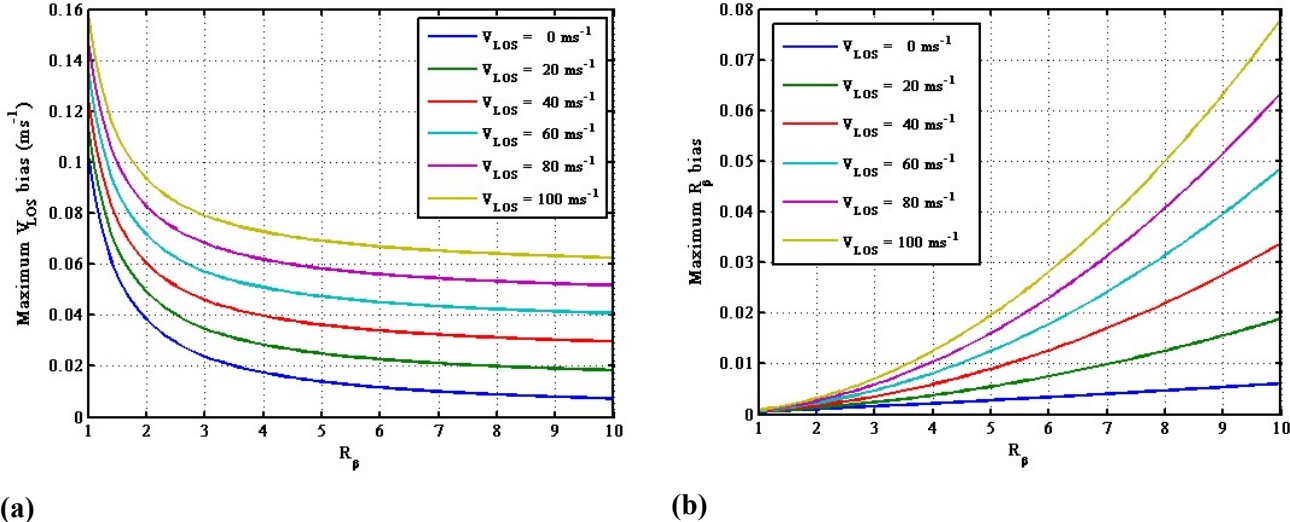

**(a)**                                                                          **(b)**

**Figure 12 Maximum measurement bias for a $10^{-3}$ relative calibration error in the sensitivity $a_i$ of one QMZ channel, as a function of the scattering ratio (a) LOS velocity error, (b) scattering ratio relative error for different LOS velocities (VLOS).**

The sensitivity parameters $a_i$ and modulation parameters $M_i$ are determined by the calibration sequence presented in Sect. 3.5. As tested on the HSRD-LNG QMZ, fits of the recorded signals with the model described by Eq. (1) are obtained with a correlation better than 99.5 % (Bruneau et al., 2015, see also Appendix C). This allows determining the $a_i$ and $M_i$ parameters with a high accuracy.

## 5.       Discussion

From parameters reported in Table 2, and results presented in the previous section, the performance in the determination of the HLOS wind speed can be optimized using a single compact QMZ. This MZI design offers a high value alternative to the whole cascade FP-DE and Fizeau interferometric system implemented in the present Aeolus design. A better overall performance is expected to be achieved as based on realistic parameters derived from airborne operation with a minimized risk and increased design compactness and reliability. As compared to the 35° LOS angle of Aeolus, a higher 45° slant viewing

allows to increase the LOS horizontal projection and also to get the same vertical resolution with a larger range gate and hence collected signal, slightly increasing performance, despite a lower atmospheric transmission and larger range.

The expression of errors has been compared to observations (see Appendix C), confirming the pertinence of the numerical model used. One can see from Table 3 that the simulated performance is better than the original requirement for ADM-Aeolus in the lower stratosphere and free troposphere (even keeping a 1000 m and 500 m resolution, respectively, see Fig. 8), and

slightly worse in the boundary layer for a varying vertical resolution. This does not appear critical as meteorological stations are providing measurements at the surface as well as spaceborne scatterometers over ocean. The achievable performance

appears to still be acceptable and does not justify the introduction of a second interferometer optimized for particulate scattering, which would significantly complicate the instrument.

Aeolus has been studied to be operated with a 130 mJ / 100 Hz transmitter to meet the requirements but the development of the operational system only allowed using a 65 mJ / 50 Hz transmitter. Considering equations of errors with SNR depending on the square root of the emitted power (product of energy and repetition rate) at high SNR, the reduction in the horizontal resolution (90 km instead of 50 km), the change in altitude (320 km instead of 395/425 km), ALADIN in space should produce a statistical error increased by a factor of about 3.5 as compared to its original dimensioning. The actual standard deviation of

the Aeolus measurements achieved is about 4 ms$^{-1}$ with the Rayleigh channel in the free troposphere and about 2 to 3 ms$^{-1}$ with the Mie channel in the lower two kilometers, both for a 90 km horizontal resolution (Reitebuch et al., 2019, Geiss et al., 2019, Witschas et al., 2020, Martin et al, 2021). In the troposphere the actual Aeolus performance appears to be degraded by transmission losses by a factor 1.5 to 2 with respect to simulations (Reitebuch et al., 2019). This is mostly due to the narrow field of view requiring high pointing accuracy due to the low DE-FP acceptance angle. The proposed field compensated QMZ

design offers more flexibility and its actual performance are expected to be closer to the simulations.

The instrument calibration error in the sensitivity $a_i$ on one channel by $10^{-3}$ propagates to wind measurements with a maximum LOS velocity error of 0.15 ms$^{-1}$ (0.2 ms$^{-1}$ in HLOS) depending on the interference phase (the bias averaged on all phases is negligible). This can appear relatively sensitive, but in fact the calibration process given in Section 3.5 has been validated with the HSRD-LNG airborne system and indeed allows such a calibration accuracy. Actually,

the average 0.12 ms$^{-1}$ LOS wind speed bias observed with this system on a long series of ground returns (Bruneau et al, 2015) shows that the residual phase dependent bias is very limited. To monitor system health and avoid degradation of performance with time due to ageing of optical components, the calibration operation is nevertheless to be done regularly. Note also that the high sensitivity of the wind speed measurement with regard to the instrument calibration is inherent to the spectral discrimination based on molecular scattering because of its large spectral distribution. For comparison, the characteristics of

the DE-FP used at the Observatoire de Haute Provence (Souprayen et al, 1999b) lead to a bias of 0.38 ms$^{-1}$ in the LOS velocity measurement for a $10^{-3}$ sensitivity error on one channel. The actual ALADIN HLOS wind velocity measurement bias has been estimated to be around 1.5 ms$^{-1}$ with a latitude and orbit phase dependence (Martin et al, 2021).

The wind measurement accuracy in presence of a cirrus cloud of variable optical depth, as frequently observed in the atmosphere, is not degraded so much for low cloud optical depths (Fig 12b). HLOS wind speed errors are maintained below 3

m.s$^{-1}$ and 5 m.s$^{-1}$ at altitudes of 1 km (top of the atmospheric boundary layer) and 5 km respectively in presence of cirrus with optical depths up to 0.5. Measurement errors in cirrus clouds remain low, allowing higher horizontal resolutions, which is highly valuable as they are close to the upper tropospheric jet.

Unlike the FPI-based HSRL technique proposed for ATLID, the QMZ technique does not attempt to directly separate molecular and particulate signals but delivers, in addition to the total signal $S_{tot}$, the atmospheric interference modulation $M_{atm}$

from which the scattering ratio $R_\beta$ is derived (see Appendix A). The performance of the QMZ in terms of backscatter coefficient

measurements are thus expressed differently than from those given by the EarthCare instrument. A determination of the particulate component of the signal is however possible by using molecular backscatter profiles from a model in Eq. A11, introducing small additional errors (see previous section and Appendix D). The retrieval of extinction can then be done in a very similar way as for conventional HSRL. We nevertheless can see (Fig. 9b) that the scattering ratio random error of 1 % in cirrus clouds with scattering ratios larger than 5 (with 500 m vertical and 50 km horizontal resolutions), as it is the case for most semi-transparent cirrus clouds, compares favorably with the radiometric errors of 50 % and 15 % for particulate and molecular signals respectively required for EarthCare (for 100 m / 300 m vertical and 10 km horizontal resolutions - Hélière et al, 2012). Starting from the scattering ratio (as well as from separated molecular and particulate signals) it is further needed to use an atmospheric model to derive the particulate backscatter and extinction coefficients coefficient (Eq. A12, A13). An error of about 2K in the temperature leads to an error smaller than 1 % for both $\beta_{par}$ and $\alpha_{par}$ in the PBL. These errors account for the molecular density, as for conventional HSRL, and for the $M_{mol}$ temperature dependence which can be seen as the equivalent of the temperature dependence of the transmission of the molecular channel in conventional technique. The QMZ performance simulations have been conducted with a vertical resolution derived from the Aeolus requirements and A-CCD characteristics taken from the ALADIN design. New A-CCDs with shorther time gates as those planned for ATLID would result in a higher vertical resolution compatible with the EarthCare requirements. Nevertheless, as the proposed configuration is aiming as a first priority to high performance wind measurements, it is first requested to keep a low read noise per range gate as compared to the useful signal. Reducing range gate duration thus implies to simultaneously reduce read noise to keep an accuracy pertinent to the survey of dynamics and cloud parameters at the global scale.

In addition, the QMZ measurement is performed with a reference taken from the emitted pulse and does not require any frequency locking of the laser or the interferometer. The instrument is also tolerant to transmitter frequency linewidth, jitter (200 MHz RMS during half a second, included in the performance simulations) and unaffected by long-term frequency drifts. The unfolded measurement range is $\pm$ 832 ms$^{-1}$ accepting the maximal earth rotational speed of 463 ms$^{-1}$ without the need of a latitude dependent LOS correction.

The design includes a relatively wide field angle (100 µrd, about 5 times larger than that of ALADIN, similar to ATLID and CALIOP instruments) which relaxes the requirements on the transmitter-receiver alignment. This wider aperture angle allows also an increase in the emission cone, and thus a reduction in the energy density at the surface, potentially also allowing the emission of other wavelengths in eye safety conditions. The solar background is however kept at the same level due a more severe interference filtering.

## 6.     Conclusion

The Aeolus mission has emphasized new capabilities of lidar in space for wind measurements. The discussion of the Aeolus follow-on program allows to revisit some of the choices made that have revealed technical constraints limiting achievable performance. In this paper we explore possible solutions to relax some of the constraints identified, keeping initial performance

objectives identified. A new spaceborne lidar design for the horizontal wind speed measurement using a single QMZ interferometer technique is proposed for consideration for operational missions following Aeolus and EarthCare, taking advantage of advances made for them both. The changes in the payload are minimized as the lidar system would use the same telescope aperture as Aeolus and a comparable 355-nm transmitter. A single Quadri-channel Mach-Zehnder interferometer (QMZ) is selected to replace the cascade double-edge Fabry Perot (DE-FP) and Fizeau interferometers. The design of the QMZ can provide robustness and compactness, and can be developed with molecular adherence as for Aeolus interferometers. The QMZ offers a large acceptance angle reducing mechanical and thermal constraints. These choices allow to overcome the limitations induced by the need of a narrow field of view for the DE-FP and the use of a high accuracy alignment control and/or a drastic reduction in thermo-mechanical constraints. The QMZ design also relaxes constraints on the telescope focusing. There is no bias induced by aerosol scattering on Doppler shift, which also represents a strong advantage for the needed bias corrections in the assimilation process. The instrument is optimized for wind measurements over the whole troposphere as for Aeolus. It also authorizes the retrieval of cloud and aerosol optical properties with a satisfying accuracy as compared to the EarthCare mission. It further allows the implementation of a depolarization channel as for ATLID and HSRD-LNG, without degrading overall performance and offering unbiased operation. For the retrieval of aerosol and cloud backscatter properties, the necessary knowledge of the molecular spectrum (including Brillouin scattering) can be derived from meteorological analyses to provide products with the required accuracy.

The performance study shows that a statistical error on the horizontal wind speed better than 2 ms$^{-1}$ can be achieved, from the boundary layer up to the tropopause for a 500 m vertical resolution and a 50 km horizontal resolution. Accordingly, the measurement bias can be as small as 0.2 ms$^{-1}$, provided a proper calibration is done. In addition to wind speed, the instrument can retrieve the scattering ratio with a relative random error of 1 to 2 % over most of the troposphere, well matched to high cloud study and radiatively significant aerosol load. The performance analysis on the QMZ interferometer is supported by the overall performance of the UV HSRD-LNG airborne lidar developed and operated by LATMOS and INSU.

**Appendix A. Signal processing and noise dependent measurement errors**

**Signal processing**

In this appendix we refer to BP03 to recall expressions of errors of atmospheric wind speed and attenuated backscatter coefficients using a QMZ interferometer. Let us start from the optical lidar signal $S_{atm}$ (in number of photons) at the QMZ input as

$$S_{atm}(R) = S_{mol}(R) + S_{par}(R) = \frac{EAT_{inst}}{h\nu} \int_{r=R-\delta R/2}^{r=R+\delta R/2} \frac{(\beta_{mol}(r)+\beta_{par}(r))}{r^2} T_{atm}^2(r)dr \qquad (A1)$$

where $E$ is the emitted energy, $A$ the telescope area, $T_{inst}$ the instrumental (emission and reception) transmission, $h\nu$ is the emission photon energy, $R$ is the range, $T_{atm}(r) = exp\left[-\int_0^r (\alpha_{mol}(r') + \alpha_{par}(r'))dr'\right]$ and the mol and par subscripts are related to the molecular and particulate scattering respectively.

The signal delivered by each channel of the A-CCD outputs (in photoelectrons), for $i = 1$ to 4, can be written

$$S_i = \frac{S_{tot}}{4} a_i \left[1 + M_i M_{atm} sin\left(\varphi + (i-1)\frac{\pi}{2}\right)\right] + S_{b,i} \qquad (A2)$$

$a_i$ is the relative photometric efficiency of channel $i$ with $\sum_{i=1}^4 a_i = 4$, $M_i$ is the intrinsic modulation of channel $i$. $S_{b,i}$ is the background signal due to solar detected light and A-CCD intrinsic noise.

As all the photons incident on the MZI arrive on the detectors, we have:

$$S_{tot} = \frac{4}{\sum_{i=1}^4 M_i^{-1}} \sum_{i=1}^4 \frac{(S_i - S_{b,i})}{a_i M_i} = \eta S_{atm} \qquad (A3)$$

where $\eta$ is the mean quantum efficiency of the detectors.

$\varphi$ is the interference phase and $M_{atm}$ is the atmospheric effective interference modulation given by the molecular and particulate atmospheric backscattered signals as

$$M_{atm} = \frac{M_{par}(R_\beta - 1) + M_{mol}}{R_\beta} \qquad (A4)$$

where the lidar scattering ratio $R_\beta$ is defined by the ratio of the total to the molecular backscatter coefficients as $R_\beta = \frac{\beta_{mol} + \beta_{par}}{\beta_{mol}}$. Depending on the scattering ratio, the atmospheric modulation coefficient $M_{atm}$ varies between $M_{mol}$ and $M_{par}$, the interference modulations given by pure molecular and particulate signals respectively.

After subtraction of the background, the four signals are combined two-by-two in order to produce a complex signal $Q$ (with in-phase and quadrature components):

$$Q = Q_2 + iQ_1 \quad \text{with} \quad Q_1 = \frac{a_3(S_1 - S_{b1}) - a_1(S_3 - S_{b3})}{a_3 M_3(S_1 - S_{b1}) + a_1 M_1(S_3 - S_{b3})}; Q_2 = \frac{a_4(S_2 - S_{b2}) - a_2(S_4 - S_{b4})}{a_4 M_4(S_2 - S_{b2}) + a_2 M_2(S_4 - S_{b4})} \qquad (A5)$$

The interference phase $\varphi$, is obtained by the argument of the complex signal $Q$:

$$\varphi = arg(Q) \tag{A6}$$

Subtracting the reference phase $\varphi_r$, obtained by the same way on a highly attenuated pick-up of the laser emission one can obtain the line-of-sight (LOS) particles velocity VLOS with:

$$V_{LOS} = \frac{c\lambda}{4\pi\Delta}(\varphi - \varphi_r) \tag{A7}$$

where $\Delta$ is the MZI optical path difference, $\lambda$ the operating wavelength and $c$ the light celerity both in vacuum. Using this differential approach $V_{LOS}$ can be obtained on the whole measurement range $\pm\frac{c\lambda}{4\Delta}$ without the need to lock the emitting frequency with the interference phase.

The atmospheric modulation is obtained by the modulus of $Q$ divided by the modulus of the reference signal:

$$M_{atm} = \frac{|Q|}{|Q_r|} \tag{A8}$$

One can see, from Eq. A3 that, once $M_{mol}$ and $M_{par}$ are determined, $M_{atm}$ is giving access to the scattering ratio $R_\beta$.

$$R_\beta = \frac{M_{par} - M_{mol}}{M_{par} - M_{atm}} \tag{A9}$$

Separated molecular and particulate signals can be obtained (with the same instrumental constant) using:

$$S_{mol} = \frac{M_{par} - M_{atm}}{M_{par} - M_{mol}}\frac{S_{tot}}{\eta} \quad ; \quad S_{par} = \frac{M_{atm} - M_{mol}}{M_{par} - M_{mol}}\frac{S_{tot}}{\eta} \tag{A10}$$

but this step is not necessary for the retrieval of the particulate backscatter and extinction coefficients obtained as follows:

$$\beta_{par} = \beta_{mol}(R_\beta - 1) = \beta_{mol}\frac{M_{atm} - M_{mol}}{M_{par} - M_{atm}} \tag{A11}$$

The total particulate optical depth over the vertical column can be derived from the total signal and the scattering ratio

$$\tau_{par} = \frac{1}{2}Ln\left(\frac{R_\beta \cdot \beta_{mol}}{r^2 S_{tot}}\right) - \tau_{mol} \tag{A12}$$

One way to determine the particulate extinction coefficient is to derive equation A12 giving the particulate optical depth with
altitude z as

$$\alpha_{par} = \frac{1}{2}\frac{d}{dr}\left(Ln\left(\frac{R_\beta \cdot \beta_{mol}}{r^2 S_{tot}}\right)\right) - \alpha_{mol} \tag{A13}$$

which allows to get rid of calibration. This method is sensitive to noise, and other ways to derive extinction can provide better results. We will however use this conservative approach for the sake of simplicity.

**Preliminary evaluation of M_par and M_mol**

Assuming Gaussian spectral profiles, the two interference modulations $M_{mol}$ and $M_{par}$ can be expressed as a function of the optical path difference $\Delta$ and the 1/e linewidth $\gamma$ (expressed in wavenumber) of the related spectral functions of atmospheric scattering convolved by the laser emitted width, as given in BP03, so that

$$M_s = exp[-(\pi\gamma_s\Delta)^2] \qquad \text{where } s \text{ stands for } mol \text{ or } par \qquad (A14)$$

For the particle scattering, $\gamma_{par}$ is defined as a first approximation by the laser source linewidth $\gamma_{las}$. Assuming $\gamma_{las}$ on the order
of $3.10^{-3}$ cm$^{-1}$ (or 100 MHz, corresponding to the transform limit of a 3 ns pulse), we see that we obtain $\gamma_{par}\Delta \approx 10^{-2}$ and $M_{par}\approx$ 1 for an OPD value of 3 cm. We thus remain in the case of a high contrast for the particulate signal, with some margin on the laser linewidth.

For the molecular scattering, $\gamma_{mol}$ is dominated by the thermal molecular velocity:

$$\gamma_{mol} = \frac{2}{\lambda c}\sqrt{\frac{2kT}{m}} \qquad (A15)$$

and is about $7.10^{-2}$cm$^{-1}$, which implies that $\gamma_{mol}.\Delta \approx 0.2$ and $M_{mol} \approx 0.6$ for an OPD of 3 cm.

**Noise dependent statistical error**

We give here the random error depending on the detection noise. Assuming, for simplicity, that all the relative sensitivities $a_i$ are equal to 1 and that all the instrumental modulations $M_i$ are equal to $M_0$, the standard deviation of $V_{LOS}$ is given in BP03 as

$$\sigma(V_{LOS}) = \frac{c\lambda}{4\pi\Delta} \frac{\sqrt{2}}{SNR} \frac{1}{M_0 M_{atm}} \sqrt{1 - \frac{1}{2}F_B M_0^2 M_{atm}^2 sin^2(2\varphi)} \qquad (A16)$$

where $SNR$ is the signal-to-noise ratio and $F_B$ is a correlation coefficient between the four detection channels given by $F_B = \frac{S_{tot}-S_b}{S_{tot}+S_b}$ where $S_{tot}$ is the number of signal photo-electrons and $S_b$ is the total number of photo-electrons of the radiative and detection background (both summed with the ponderation given in Eq. A3). We also assume here that the background can be measured over an extended range gate and subtracted for any measurement pixel with a negligible impact on bias and SNR. This assumes that the background noise is taken with a much higher SNR than the atmospheric signal. For accurate
measurements, the SNR needs to be high so we have $S_{tot} \gg S_b$ and $F_B \approx 1$ leading to a minimum square-root factor.

If, as we propose, the laser frequency is not locked to the interferometer, the phase can take any value between 0 and $2\pi$. For the performance assessment of the instrument, we thus average the error on $\varphi$ and obtain:

$$\sigma(V_{LOS}) = \frac{c\lambda}{4\pi\Delta} \frac{\sqrt{2}}{SNR} \frac{1}{M_0 M_{atm}} \sqrt{1 - \frac{M_0^2 M_{atm}^2}{4}} \qquad (A17)$$

The factor $D_c = \frac{1}{M_0 M_{atm}} \sqrt{1 - \frac{M_0^2 M_{atm}^2}{4}}$ can be seen as a degradation factor on the wind error due to the contrast degradation.

For high scattering ratios we have $M_{atm} \approx M_{par} \approx 1$, and assuming a perfect MZI, we have $M_0 \sim 1$, so that $D_c = 1$. In clear air, for which we have $M_{atm} \approx M_{mol} \approx 0.6$, the degradation factor is $D_c = 1.6$. This is intrinsic to the QMZ technique, for which error is reduced in presence of particle scattering.

The relative standard deviation of the statistical error on $R_\beta$ is linked to the error on modulus of $Q$ (Eq. A8). After averaging over $\varphi$, it can be expressed as (BP03)

$$\frac{\sigma(R_\beta)}{R_\beta} = \frac{\sqrt{2}}{SNR} \frac{R_\beta}{M_0(M_{par} - M_{mol})} \sqrt{1 - \frac{3}{4} F_B M_0^2 M_{atm}^2} \qquad (A18)$$

We can then derive the error on $\beta_{par}$:

$$\sigma(\beta_{par}) = \frac{\sqrt{2}}{SNR} \frac{R_\beta(\beta_{par} + \beta_{mol})}{M_0(M_{par} - M_{mol})} \sqrt{1 - \frac{3}{4} F_B M_0^2 M_{atm}^2} \qquad (A19)$$

The error on the particulate extinction coefficient is given (after averaging over $\varphi$) by (BP03):

$$\sigma(\alpha_{par}) = \frac{1}{\sqrt{2}\delta r} \frac{1}{SNR} \sqrt{1 + \frac{2R_\beta^2}{M_0^2(M_{par} - M_{mol})^2} \left(1 + \frac{3}{4} F_B M_0^2 M_{atm}^2\right) - \frac{R_\beta M_{atm}}{M_{par} - M_{mol}}(1 + F_B)} \qquad (A20)$$

## Appendix B. MZI field compensation and aperture

A MZI produces interferences between waves propagating in two arms, one in vacuum with length $L_1$, the other in a glass of optical index $n$ and length $L_2$ (not taking into account the elements which are common to both arms). The OPD $\Delta$ between these two waves depends on the incidence angle $i$ on the interferometer:

$$\Delta(i) = \frac{nL_2}{\cos(r)} - \frac{L_1}{\cos(i)} \quad \text{with} \quad \sin(i) = n \sin(r) \qquad (B1)$$

After mathematical developments we obtain:

$$\Delta(i) = nL_2 - L_1 + \frac{1}{2}\left(\frac{L_2}{n} - L_1\right)\sin^2(i) + \frac{3}{8}\left(\frac{L_2}{n^3} - L_1\right)\sin^4(i) + \varepsilon \qquad (B2)$$

where $\varepsilon$ represents the terms of even order superior to 4 in $\sin(i)$.

The field compensation is obtained for $L_2 = nL_1$ which cancels the term in $\sin^2(i)$. The remaining angle dependence can be written at the fourth order on i:

$$\Delta(i) = \Delta_0 \left(1 - \frac{3}{8n^2} i^4\right) \quad \text{with} \quad \Delta_0 = \Delta(i = 0) = nL_2 - L_1 \qquad (B3)$$

when averaged over the MZI aperture between 0 and $i_{max}$ we have:

$$\bar{\Delta} = \frac{2}{i_{max}^2} \int_0^{i_{max}} \Delta(i) i\, di = \Delta_0 \left(1 - \frac{i_{max}^4}{8n^2}\right) \qquad (B4)$$

$$\overline{\Delta^2} = \frac{2}{i_{max}^2} \int_0^{i_{max}} \Delta^2(i)idi = \Delta_0^2 \left(1 - \frac{i_{max}^4}{4n^2} + \frac{9}{5}\frac{i_{max}^8}{2^6 n^4}\right) \tag{B5}$$

The residual RMS wavefront error after field compensation is then:

$$\sigma_{RMS} = \sqrt{\overline{\Delta^2} - \bar{\Delta}^2} = \Delta_0 \frac{i_{max}^4}{4\sqrt{5}n^2} \tag{B6}$$

As the beam étendue is maintained over its propagation through the instrument, the product of the useful interferometer aperture $D_{MZI}$ by the apparent source field angle $\alpha_{MZI}$ is equal to the product of the receiving telescope aperture $D_{tel}$ by its field-of-view $\alpha_{tel}$. Besides, it can be shown that the minimal aperture $D_L$ for a beam propagating on a distance $L$ with a specified $D_{tel}$ by $\alpha_{tel}$ product is $D_L = 2\sqrt{D_{tel}\alpha_{tel}L}$. As, for field compensation, the minimal optical propagation length in the MZI (both arms are equals) is $L_{min} = L_1 = \frac{\Delta_0}{n^2-1}$. We can therefore consider that a realistic MZI (including all the elements common to the two arms) can be built with an internal propagation length of $2\Delta_0$. The resulting useful diameter of the MZI is then:

$$D_{MZI} = 2\sqrt{2D_{tel}\alpha_{tel}\Delta_0} \tag{B7}$$

Then, for the instrumental parameters of this study, $D_{tel} = 1.5$ m, $\alpha_{tel} = 0.1$ mrd and OPD = 3.2 cm, the useful MZI aperture is $D_{MZI} = 6.2$ mm.

Using Eq. B6 and B7 with $i_{max} = \frac{\alpha_{MZI}}{2} = \frac{D_{tel}\alpha_{tel}}{2D_{MZI}} = \frac{\sqrt{D_{tel}\alpha_{tel}}}{4\sqrt{2\Delta_0}}$ , we obtain:

$$\sigma_{RMS} = \frac{D_{tel}^2\alpha_{tel}^2}{2^{12}\sqrt{5}n^2\Delta_0} \tag{B8}$$

With our instrumental parameters we find a residual wavefront distortion of $\sigma_{RMS} = 3.7 \ 10^{-2}$ nm due to the field compensation.

**Appendix C. Analysis of observed HSRD-LNG performance**

**Wind measurements**

In addition to the results presented in Bruneau et al, 2015, we report here wind measurements performed with HSRD-LNG aboard the SAFIRE Falcon 20 aircraft during the NAWDEX-EPATAN field experiment (Schäfler et al, 2018) on October 4th 2016.

We remind that due to its OPD of 20 cm the HSRD-LND interferometer is able to measure wind only from the particle scattering. The measurements were therefore performed in extended cirrus clouds from an altitude of 12 km, during 360° turns of the aircraft at a constant roll angle of 27°, providing several LOS measurements at various levels in the cloud, allowing performance of a velocity azimuth display (VAD) analysis at each level. For each VAD, the average wind module and direction is computed from a sine fit on approximately 100 LOS measurements of 100 shots each, and extends on a circle of approximately 20 km in diameter for a total duration of 500 s. Figure C1a shows an example of measurements and VAD sine fit at an altitude of 7.9 km with a vertical resolution of 100 m.

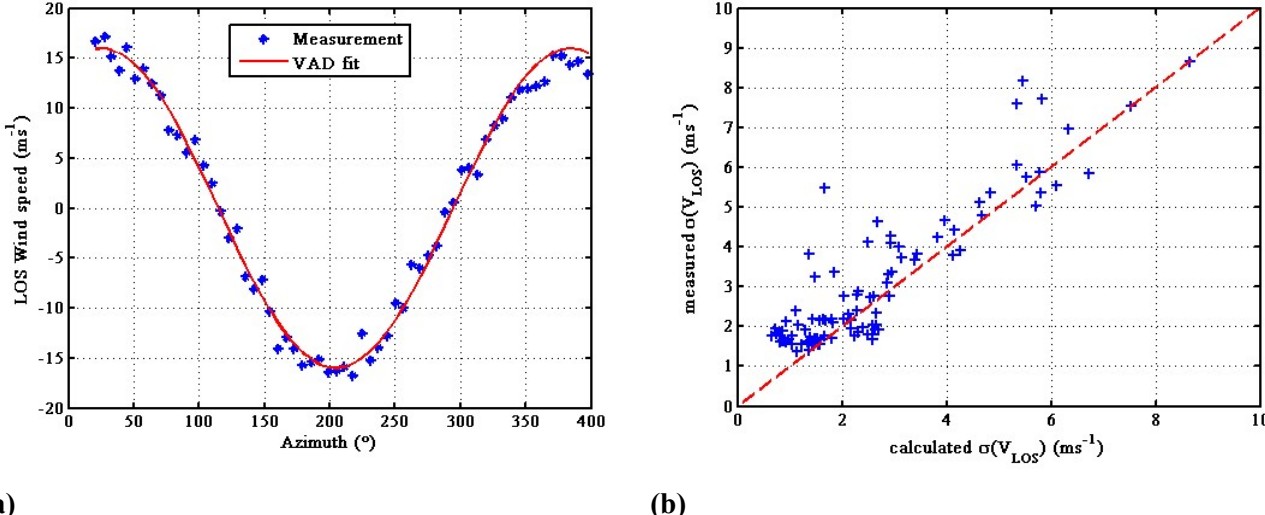

(a)                                                              (b)

**Figure C1 a) LOS wind speed retrieved by HSRD-LNG over 100 shots at an altitude of 7.9 km b) Comparison of the measured and calculated LOS wind speed ($V_{LOS}$) errors**

Taking the sine fit as reference, we can compute the standard deviation of the LOS measurements and compare it to the standard deviation calculated with Eq. A17. Figure C1b shows the comparison of the measured and calculated LOS wind speed standard
deviations computed at different altitudes on three different VAD measurements taken at 16:37, 17:34 and 18:00 UTC (for a total of 98 measurements). Note that because of the spatial and temporal extents of the VADs, the measured standard deviations include a part of natural variability and are in the average slightly higher than the calculated standard deviations. Taking into account the natural variability, we can see that the measured and calculated errors are in good agreement and then consider that Eq. A17 is validated.
To further illustrate the capacity of the airborne HSRD-LNG to measure wind, Fig. C2 a, b and c present the three wind profiles retrieved from VADs analysis on October, 10th, 2016, compared to collocated radiosonde measurements. Figure C2d presents the histogram of the wind speed and wind direction differences between VAD lidar and dropsonde profiles. Table C1 summarizes the horizontal wind speed $V_{HOR}$ and wind direction $V_{DIR}$ differences between lidar and dropsonde profiles and the related standard deviations compared to the mean calculated standard deviations, for the 3 cases of October 10. As the lidar
and dropsonde profiles cannot be perfectly collocated the mean standard deviation of the profiles difference includes the wind natural variability with different integrations in time and space, but differences remain small.

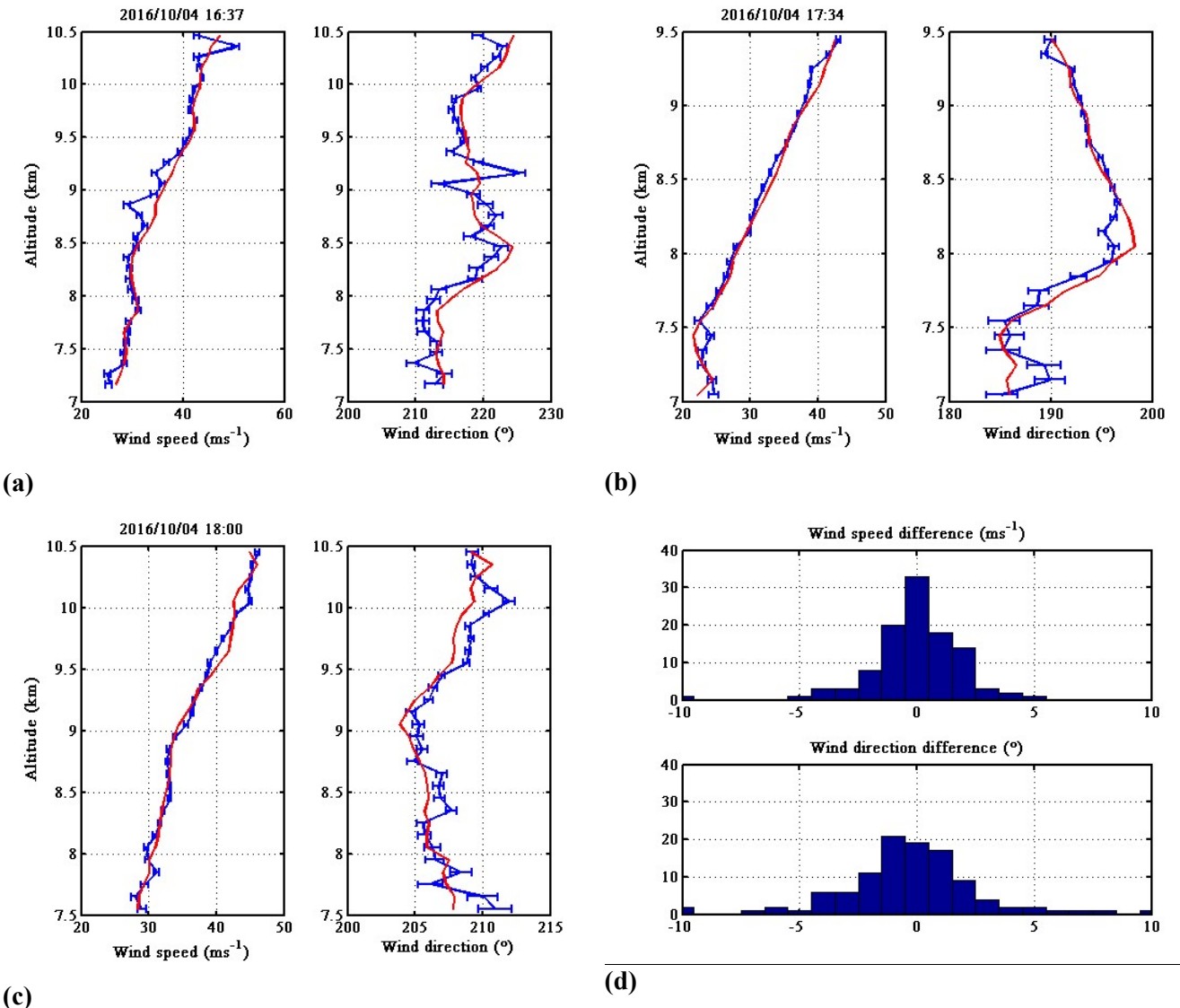

**(a)**

**(b)**

**(c)**

**(d)**

Figure C2 HSRD-LNG VAD wind profiles (blue) with calculated error bars (standard deviation) and collocated radiosonde measurements (red) for NAWDEX-EPATAN VADs performed on October, 4, 2016 for three dropsondes lunched at (a) 16:30 UTC, (b) 17:30 UTC and (c) 18 UTC. (d) histogram of differences of HLOS wind speeds between HSRD-LNG and related dropsonde measurements over the three vertical profiles

Table C1 Summary of mean differences between wind lidar and radiosonde profiles and related standard deviations (std).

| Profile | a | b | c | Average |
|---|---|---|---|---|
| LNG-DS Time difference (hour) | 0.06 | 0.07 | 0.03 | 0.05 |
| LNG-DS Distance (km) | 16 | 14 | 5 | 12 |
| LNG-DS mean $V_{HOR}$ difference (ms$^{-1}$) | -0.81 | -0.11 | 0.09 | -0.28 |
| LNG-DS mean $V_{DIR}$ difference (°) | -1.1 | -0.2 | 0.8 | -0.17 |
| LNG-DS std $V_{HOR}$ difference (ms$^{-1}$) | 1.72 | 1.01 | 0.92 | 1.22 |
| LNG-DS std $V_{DIR}$ difference (°) | 2.4 | 1.5 | 1.1 | 1.67 |
| LNG mean calculated std $V_{HOR}$ (ms$^{-1}$) | 0.98 | 0.83 | 0.59 | 0.80 |
| LNG mean calculated std $V_{DIR}$ (°) | 1.74 | 1.8 | 1.0 | 1.51 |

**Particle backscatter coefficient measurements**

To assess the measurement error on the particle backscatter coefficient, we compiled several ground-based measurements in the boundary layer. The raw $\beta_{par}$ measurement variability includes both the instrumental error and the natural variability. In order to separate the two terms we used a spectral analysis of the temporal series for each vertical range gate such as developed in Frehlich et al, 1998. The instrumental error is given by the plateau of the spectrum at high frequencies. These measured

errors are compared to the calculated errors obtained from Eq. A19 in Fig C3. As measured and calculated errors are in fairly good agreement we can consider that the error calculation used in the performance model for the backscatter coefficient is validated.

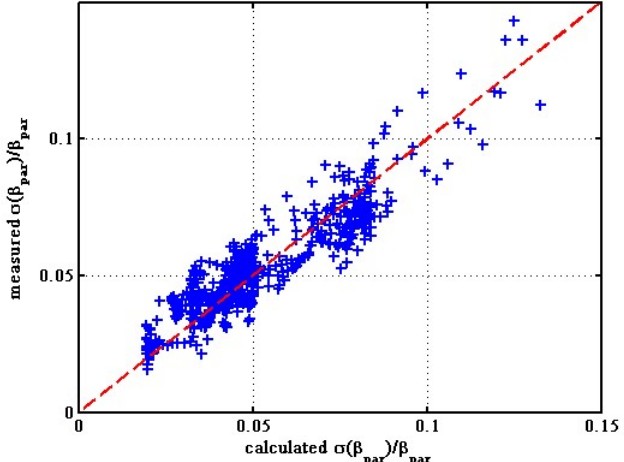

**Figure C3 Comparison of measured and calculated relative errors (standard deviation) on the particle backscatter**
**coefficient.**

## Appendix D. Atmosphere-modelling dependent errors in the retrieval of particulate backscatter and extinction coefficients

As already pointed out, the wind speed measurement does not require any atmospheric modeling (unlike DE-FP operation) since the interference phase determination is independent of the actual interference modulation $M_{atm}$. Nevertheless, the particle backscatter coefficient and extinction retrieval (Eq. A8 –A10) requires the knowledge of $\beta_{mol}$. $M_{mol}$ can be obtained with some atmosphere modeling and assumptions.

The calculation of $M_{mol}$ is not as straightforward as that given by Eq. A14 since it must include the effect of Brillouin scattering. For an operating wavelength of 355 nm the Brillouin doublet separation is much smaller than the thermal linewidth. The resulting spectrum is a broadening and distortion of the Gaussian profile. A correction can be derived from a model (Tenti et al., 1974) or measurements (Witschas et al., 2010). The effect is roughly proportional to the pressure with a maximum relative broadening of 1.7 % near ground. This broadening causes a relative decrease of $M_{mol}$ of up to 6 $10^{-3}$ as compared to a Gaussian thermal line (Fig C1). The Brillouin effect must be included in the calculation of $M_{mol}$ but the uncertainty on the actual atmospheric pressure leads to a second-order variation of $M_{mol}$ that can be neglected. This constraint is less critical here for the quantification of the molecular backscatter coefficient than for wind measurements using a DE-FP device (Dabas et al., 2008).

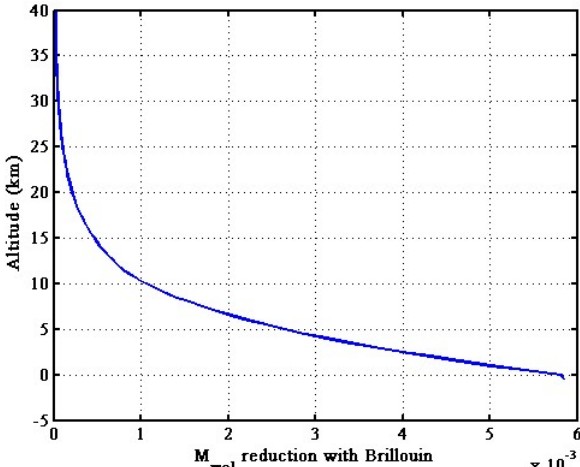

**Figure D1 Reduction of $M_{mol}$ caused by the Brillouin effect**

In contrast, the dependence of $M_{mol}$ on temperature, given in Fig D2a for a standard profile, must be taken into account. The molecular linewidth $\gamma_{mol}$ varies as the square root of temperature (see Eq. A15), which leads to a variation of about 20 % over the atmospheric column. The knowledge of atmospheric temperature to within 2K allows reducing uncertainty on $M_{mol}$ to less than 1 %.

The other source of error on the particle backscatter coefficient comes from the modeling of the molecular backscatter coefficients which is proportional to the molecular density and then depends on the temperature vertical profile. The sensitivity of the relative error with regard to temperature is presented on Fig. D2b.

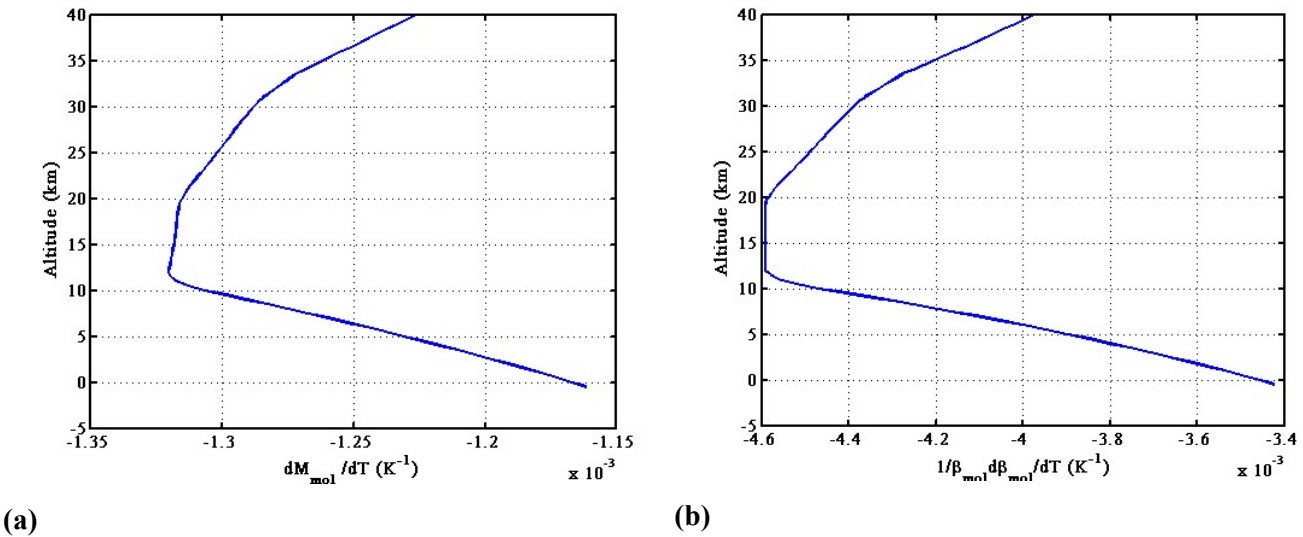

(a)                                                                                    (b)

**Fig D2 Temperature sensitivity for (a) $M_{mol}$ and (b) $\beta_{mol}$ as a function of altitude.**

The relative error on the retrieval of the particle backscatter coefficient caused by atmospheric analysis errors is derived from Eq. A11:

$$\left[\frac{\sigma(\beta_{par})}{\beta_{par}}\right]_{atm} = \left[\left(\frac{d(\beta_{mol})}{\beta_{mol}dT}\right)^2 + \left(\frac{d(M_{mol})}{(M_{par}-M_{mol})dT}\frac{1}{R_\beta-1}\right)^2 - \frac{2}{(M_{par}-M_{mol})(R_\beta-1)}\left(\frac{d\beta_{mol}}{\beta_{mol}dT}\frac{dM_{mol}}{dT}\right)\right]^{\frac{1}{2}}\sigma(T) \qquad (D1)$$

Considering Eq. A13 and assuming that the temperature errors between range gates are independent and with the same standard deviation $\sigma(T)$, we obtain for the error on the particle extinction coefficient caused by the atmosphere modelling uncertainties:

$$\left[\sigma(\alpha_{par})\right]_{atm} = \left[\left(\frac{cos(\theta)}{\sqrt{2}\Delta z}\frac{d(\beta_{mol})}{\beta_{mol}\,dT}\right)^2 + \left(\frac{cos(\theta)}{\sqrt{2}\Delta z}\frac{d(M_{mol})}{(M_{par}-M_{mol})dT}\right)^2 - \frac{2}{(M_{par}-M_{mol})}\left(\frac{cos(\theta)}{\sqrt{2}\Delta z}\right)^2\frac{d(\beta_{mol})}{\beta_{mol}dT}\frac{d(M_{mol})}{dT} + \left(\frac{8\pi}{3}\frac{d(\beta_{mol})}{dT}\right)^2\right]^{\frac{1}{2}}\sigma(T) \qquad (D2)$$

where $\Delta z$ is the vertical resolution and $\theta$ the LOS nadir angle.

*Acknowledgements*

The authors wish to acknowledge the CNES for their support in developing the airborne instrument HSRD-LNG and in related field experiments. ESA is also acknowledged for their support in the NAWDEX-EPATAN field experiment in 2016. The authors would like to warmly thank all our colleagues from LATMOS and INSU-DT for their contributions which made the HSRD-LNG project a reality.

The two anonymous reviewers are acknowledged for their comments and suggestions that led to the improvement of this paper. This paper is dedicated to the memory of our colleague Pierre H. Flamant, a major actor in the development of wind lidar. He passed away end of June 2020. He was the president of the early Aeolus Mission Advisory Group at ESA and contributed to several projects and field experiments paving the way in this new research area.

*Authors contributions*: JP is at the origin of the paper and has established the main issues. DB defined the instrumental design and established the performance modeling. Both authors participated in the writing of the paper.

*Competing interests:* The authors declare they have no conflict of interest.

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
