# Peer review of "A new lidar design for operational atmospheric wind and cloud / aerosol survey from space"

_Atmospheric Measurement Techniques, 2020_

## Referee Comment (RC1) · Anonymous Referee #1 · 10 Jan 2021

*Review report for manuscript amt-2020-487*

**A new lidar design for operational atmospheric wind and cloud/aerosol survey from space**

*Didier Bruneau and Jacques Pelon*

General comments:

The manuscript by Bruneau and Pelon reports a novel spaceborne wind lidar scheme for potential Aeolus follow-on missions. The proposed wind measurement principle relies on a single Quadri-channel Mach-Zehnder (QMZ) interferometer which allows for relaxed operational constraints and reduced measurement bias compared to the receiver concept based on Fizeau and Fabry-Perot interferometers that is currently implemented in the Aeolus instrument. In addition, the new design offers the capability to retrieve cloud and aerosol properties. The article discusses the optimization of the key system parameters of the QMZ and provides a simulation of the systematic and random errors, demonstrating that they are compliant with the Aeolus mission requirements.

The direct-detection wind lidar technique based on a QMZ interferometer was already described in several previous publications, primarily by the two co-authors and colleagues. In the present manuscript the concept is revisited in the context of Aeolus and its mission requirements. In this respect, the study contains novel aspects and is certainly of interest to many readers in the space-based remote sensing and Earth observation communities, particularly given the current discussions about potential Aeolus follow-on missions. The results of the simulations that were carried out to estimate the performance of horizontal wind speed and particle backscatter and extinction measurements under different atmospheric conditions are conclusive. However, a further substantiation of the performance estimations based on atmospheric measurements from the airborne LNG instrument which also relies on the proposed design is only briefly indicated and should be elaborated in more detail (see specific comments below). The manuscript is well-organized and carefully written. The derivation of physical relationships used in the simulations are concisely presented in the appendices. Overall, the quality of the manuscript is very good and, upon addressing my comments, I recommend it for publication in AMT.

Specific comments:

1. An article by Stoffelen et al. (Bull. Amer. Meteor. Soc., 101, 2020) on the requirements and capabilities of future satellite-borne wind observations was recently published. As the study, amongst others, discusses the way forward toward an operational wind profiling mission, its conclusions should also be considered and referenced in the present manuscript.
2. In the entire paper and especially in the introduction, I am also missing a reference to the Optical Autocovariance Wind Lidar (OAWL) developed by Ball Aerospace (Tucker et al., J. Atmos. Ocean. Tech., 35, 2018), although the citation is included in the reference list. The OAWL instrument also relies on a quadrature Mach–Zehnder interferometer and thus represents a very similar concept as the one described in the manuscript. Therefore, I suggest expounding on the advantages and disadvantages of the proposed lidar design compared to the OAWL system.
3. Page 5, line 152: Please clarify what is meant by off-axis MI interferometer. Otherwise many readers might be lost in the following section.

4. Page 7, lines 202 ff.: The proposed design envisages the use of multimode fibers for guiding the signal and reference beams to the receiver. What is the expected signal loss introduced by the fibers? Do the authors expect speckle noise to be an issue for the measurement accuracy? How is the mode scrambler realized and what is its expected transmission based on the experience with the LNG?

5. Although the theoretical error estimation yields convincing results, it would be desirable if the authors could add some experimental results from the LNG to support their theoretical findings. Is it possible to transfer the results from Bruneau and Pelon (2015), particularly in terms of the SNR and the impact of solar background, to the proposed spaceborne lidar in order to estimate the systematic and random wind and backscatter ratio errors based on actual atmospheric measurements? In the discussion chapter, the authors state that "a better overall performance is expected to be achieved as based on realistic parameters derived from airborne operation with a minimized risk and increased design compactness and reliability". At the end of the conclusion chapter, it says "The performance analysis on the QMZ interferometer is supported by measurements performed in the frame of the UV HSRD-LNG airborne lidar developed and operated by LATMOS." Could they please elaborate on these statements?

6. The quality of the figures should be improved. In particular, the resolution of Figs. 3 to 11 is rather low which makes it difficult for the reader to get the details of the plots.

Technical corrections:

1. Abstract: "ADM-Aeolus was successfully launched in 2018, after the technical issues raised for the lidar development **had** been solved, […]"

2. Abstract: "The study of the random **and** systematic errors arising from the uncertainties […]".

3. Page 2, line 39: "The feasibility of such heterodyne **systems** operating on particle scattering was discussed for long in the community […]".

4. "For the Aeolus mission, a technique combining two interferometers in cascade, **one matched to the narrow aerosol spectrum (Mie channel), the other matched to the broad molecular spectrum (Rayleigh channel)** was chosen to be implemented in the space-borne Atmospheric LAser Doppler INstrument (ALADIN) […]". The backscattered radiation is first directed to the Mie and then to the Rayleigh channel, readers might interpret a wrong order of the cascade.

5. Page 9 Table 2, line 3: "pla**nn**ed".

6. Page 9, line 239: HSR instead of HRS.

7. Page 10, line 244: "A quarter-wave plate is inserted in one arm of the interferometer and polarizers at its output **to** separate the signal in four interference components in phase quadrature."

8. Page 11, line 264: "Thanks to the mode scrambler these images are well defined **disks** of homogeneous illuminations."

9. Page 11, line 265: "They **are imaged onto a 16 by 8 pixels area** of the A-CCD image zone in the same way as the Rayleigh channel of ALADIN." The two spots of the ALADIN Rayleigh channel are contained in each hemisphere of the 16 x 16 image zone of the ACCD. However, it should also be noted that the spot width (FWHM) is below 2 pixels so that only a small portion of the hemisphere is illuminated.

10. "Prior to processing actual atmospheric signals, it is necessary to perform a calibration of the MZI for determining the transmission and modulation factors $a_i$ **and** $M_i$ **of** the MZI defined in Eq. (1).

11. Page 14, Eq. (3): The lower bound of the integral should read $R - \delta R/$**2**.

12. Page 18, line 445: "In presence of a cirrus cloud the wind speed accuracy is improved to a fraction of $ms^{-1}$ **due to** the increased signal return and interference modulation."

13. In general, mathematical symbols used in the main text and the Appendix should be typeset in italics in order to be conform with the style guidelines.

---

## Referee Comment (RC2) · Anonymous Referee #2 · 11 Jan 2021

Review of manuscript amt-2020-487

**A new lidar design for operational atmospheric wind and cloud aerosol survey from space**

**By Didier Bruneau & Jacques Pelon**

**Summary and General Assessment**

This paper builds on the decades of work performed and published by the authors (and their colleagues) on using Quadrature Mach Zehnder interferometry for wind and aerosol lidar systems. The paper represents an update on the QMZI approach and how it may provide an alternative to the Double Edge Fabry Perot an Fizeau spectrometers used on Aeolus, to simplify the mission, potentially without making changes in other areas. Using a radiometric model and interferometer receiver model, the authors demonstrate optimization of the system for full tropospheric coverage, resulting on a QMZ design with 3.2 cm OPD to optimize fringe contrast (and thus optimize wind retrievals) from molecular backscatter.

The authors have published data from similar QMZ systems (e.g. the LATMOS LNG airborne system) so the paper may benefit from a scaling of the performance of these systems to help demonstrate and further justify the potential for space-based operation.

The paper points out important advantages of the QMZ approach (e.g., a reduced need for frequency stability, high efficiency/optical throughput, and high angular acceptance/field of view) and its potential value for future wind mission systems. With minor edits, as listed below, I recommend this paper for publication in AMT.

**General Editorial Suggestions**

The paper has some confusing wording and sentence structure and would benefit greatly from editing by a native English speaker.

Most of the figures, at least in the version this reviewer was provided, have font that is too small or blurry to read (especially figures 3-11). Please replace with figures with higher resolution and/or larger font.

The paper presents a somewhat limited view of work done in the fields of wind lidar and HSRL. There is similar work being done in this field that the authors could use to further support and even enhance the important case that they are making in this paper. For example, presentations on OAWL at the Aeolus CalVal workshop have also supported the idea of using a short optical path difference QMZ interferometer for Aeolus and have demonstrated comparison with Aeolus data. While the interferometer implementation is different, the mathematical concept and listed advantages are quite similar.

A number of variables in the text (including italic font, subscripts, symbols) do not match those in the equations. This will presumably be caught in a more thorough editing process.

A few of the publications listed in the references section appear to be missing in the text: Baker, Benedetti, Lux, Reitebuch 2020, and Tucker. (Note, the text refers to Reitebuch 2019a, and Reitebuch 2019, but there is only one Reitebuch 2019 in the list of references. Perhaps one should be the Reitebuch 2020 reference?)

**Detailed Comments**

- Line 14: Might reword to say, "…wind profiler using a *single* fixed line-of-sight lidar from space." As some other proposed concepts have looked at fixed dual lines-of-sight.
- Line 19: "This ability is…" To what ability is the sentencing referring?
- Line 30: could clarify to say that, "… backscatter coefficients can be retrieved with uncertainties better than a few percent where backscatter levels exceed XX%, such as in the boundary layer and …."
- Lines 65-80: This is a limited list of HSRL approaches and could be expanded to include the OAWL approach as well as work being done by Fahua Shen:
    - Fahua Shen, Jie Ji, Chenbo Xie, Zhao Wang, and Bangxin Wang, "High-spectral-resolution Mie Doppler lidar based on a two-stage Fabry–Perot etalon for tropospheric wind and aerosol accurate measurement," Appl. Opt. 58, 2216-2225 (2019)
    - Tucker, S. and Weimer, C., 2018, October. Benefits of a quadrature Mach Zehnder interferometer as demonstrated in the Optical Autocovariance Wind and Lidar (OAWL) wind and aerosol measurements. In Remote Sensing of the Atmosphere, Clouds, and Precipitation VII (Vol. 10776, p. 107760E). International Society for Optics and Photonics.
- Line 123: DE-FP and Fizeau interferometers will have a "small" acceptance angle based on the finesse. Low Finesse etalons will have higher acceptance angles than higher finesse ones, but will provide poor frequency discrimination. Perhaps this can be clarified in the sentence by adding something like…, "*If they are providing high frequency discrimination*, both DE-FP and Fizeau interferometers will have small angular acceptance."
- Line 125: Don't the narrow interferometer fields of view (FOVs) *also* impose a high accuracy requirement on the alignment between the telescope and the receiver?
- Line 134: Could also include Grund 2008
    - Grund, C.J., Howell, J., Pierce, R. and Stephens, M., 2009, April. Optical autocovariance direct detection lidar for simultaneous wind, aerosol, and chemistry profiling from ground, air, and space platforms. In Advanced Environmental, Chemical, and Biological Sensing Technologies VI (Vol. 7312, p. 73120U). International Society for Optics and Photonics.
- Lines 149-153: The frequency stability is only required IF an accumulation detector is used for the observation. If a higher speed detector is used, such as a PMT or APD (e.g., CALIPSO detectors) then the reference can be updated every pulse, if needed.
- Line 161: 1.4 or sqrt(2)? Is this just for simplicity?
- Line 164-168: This may be due to the grammatical structure of the sentence, but it seems that the authors are implying that (because of the reflection from one edge of the Aeolus ALADIN DEFP is being used in the other edge) the Aeolus approach has a theoretical 1.4x advantage over the QMZI approach. However, it's unclear how it is possible to provide perfect efficiency through a double edge approach while maintaining the necessary frequency discrimination. Unlike with the QMZ approach, there will always be some molecular backscatter signal that is not allowed to pass through either edge filter and thus will be lost. I am aware the authors are quite familiar with how Aeolus (and DEFP) operates, so perhaps this was not the intent of the paragraph, but it's important to not lead other readers into an unintended comparison.
- Line 176-179: These statements could use a little bit of qualification. For example, *for short OPDs*, the particulate signal will be near unity, but not if the OPD is long compared to the laser linewidth. Perhaps just state that the discussion refers to the 3.2 cm OPD concept being presented here.

- Line 189:  The term "emitter" is used here (and elsewhere in the paper, including Table 2) however the term "transmitter" is used in much of the Aeolus literature (e.g., Reitebuch, et al., 2009).
- Line 190:  This point about "acceptance angle" (or "maximum field angle" or "range of field angles") is an important point to make!
- Line 195: Suggest the wording "…can be mounted on the same plate, *as was done on CALIPSO*, and *boresight* mirrors…."  (such language indicates heritage for the design)
- Line 220:  perhaps add:  "..to obtain a uniform illumination *and uniform field* distribution…"
- Line 225:  Also successfully implemented and tested on the airborne OAWL system (which further supports your case!)
- Line 265:  Do you mean that each spot fills an 8x8 pixel area?
- Line 261-270: The discussion is focused on using the same A-CCD as was used on Aeolus, however discussions of using a new ACCD or different type of detector are under consideration for Aeolus. Could the authors suggest what approach would be idea for a QMZ design?  Perhaps ranging from more ACCD rows, to a different type of detector?
- Table 2:  "Emitter Linewidth":  according to the text, the 200 MHz represents the allowed spectral jitter or drift over the 50 pulse (1/2 second) accumulation time and likely does not represent the laser linewidth.  Perhaps clarify this by calling it "Accumulated Emitter Linewidth (1/2 s)" or even "Allowed laser linewidth".  Also, (see above comment on "emitter" vs. "transmitter").
- Line 293:  Clarify that "*For a 3.2 cm OPD*, the modulation by the molecular return…."  For some longer OPDs, molecular return will provide NO modulation, just offset.
- Line 305:  YES, this is true, however won't there be a backscatter ratio below which the contrast difference is too small to clearly estimate?
- Line 330:  Be sure to clarify that because of the *specific type* of field compensation used in this specific design, the OPD variation is only dependent on $4^{th}$ power of the source angle.  There are other types of field compensation (e.g. cat's eye) for which the OPD variation
- Line 331: Please provide a reference for the equation provided in this line.
- Line 340:  Indeed, smaller incident angles on beamsplitters make it easier to balance reflection and transmission values.
- Line 355:  The authors suggest using the laser in multimode operation to perform the amplitude (sensitivity parameter) calibration– but for many seeded lasers, running them multi-mode comes with a risk of damage due to modal interference.
- Line 337:  modulation calibration can also be performed by allowing the laser to drift in frequency while remaining seeded.  This can be achieved fairly easily by temperature tuning the seed laser source, or laser cavity.
- Line 369-374:  Please indicate the wavelength for these profiles in the text for clarity.  Have these profiles been validated in any sense using data from Aeolus and/or CALIPSO?
- Figure 4:  Please increase the text font size and overall figure clarity.  The text is too blurry to read. Sub figures (a) and (b) do not have titles indicating the wavelength, etc.  Likewise, please indicate the wavelength for these profiles in the figure caption.
- Section 4.2:  It's nice to see this study on the impact of the horizontal light of sight angle on performance, though it would be interesting to compare it to previous studies done for Aeolus back when the mission was selected.  This type of angle analysis shouldn't be unique to this particular design, but if it is (for some reason this reviewer is missing) then please clarify.  It indicates that the difference in $1/R^2$ is less than the error in retrieving the horizontal LOS from a smaller pointing angle.

- Table 3: The authors may wish to clarify here that the total power is the same for the two systems. Aeolus requirements were based on a 2x higher laser pulse energy, and the QMZ approach is based on the as-operating Aeolus pulse energy but with a 2x pulse repetition frequency.
- Line 481: "where such an accuracy may be more acceptable than bias for assimilation purpose…" The terms "accuracy" and "bias" are often tied together. Do the authors mean that precision (or "uncertainty" or "random error") is more acceptable than bias?
- Line 508: The authors refer to a 10-3 relative error in the sensitivity calibration of each QMZ channel, however it might be good to clarify that this does (or does not) include the detector response sensitivity. For example, if one channel sees a high portion of the interferometric fringe during the outgoing pulse, will the detector response have any impact on returns for that channel?
- Line 564: the authors should clarify that "…the *short OPD* QMZ does not attempt to separate molecular…." There are long OPD QMZI designs that easily separate these two.
- Line 619-620: Is it possible to show a scaling of the performance of the UV HSRD-LNG airborne lidar to space-based operation (e.g., see Baidar et al. 2018).
- Appendix A: This is a good review of previous papers by the authors describing QMZ performance. Has data from the LNG system been used to validate this model and if so, how did they compare?
- Line 670-684: Why is 3 cm being used here for the OPD vs. the 3.2 cm chosen for the optimal system?
- Line 689: The assumption that the background can be measured over a long duration may meet challenges with highly varying cloud albedo over orbit.
- Appendix B: It's quite nice to see this analysis in a publication, showing the impact of OPD and maximum aperture driven by etendue and it would be nice to see this in the main paper, if room allows. It's important to clarify, however, that the analysis applies to QMZ structures using field compensation plates.
- Appendix C: Many of the points made in this section are really important in the argument for using a QMZ type system for an Aeolus follow-on. If possible, can more of this be moved into the main part of the document?

**Example English edits** (this is not an exhaustive list - please review further).

- Line 87: What is implied by "should take over"? Perhaps "will provide continuity for CALIPSO lidar and CloudSat radar observations for……."
- Line 98: What is meant by the phrase "….is ruled by…" ?
- Line 100: Only one reference is listed regarding the value of upper level winds to NWP, but there are many more that have been published.
- Line 107: "large efficiency detectors" should probably be "high efficiency detectors"
- Line 111: This line is awkward, perhaps the authors simply meant to say, "…the interferometer configuration has revealed operation and performance constraints."
- Line 119: "199b" should be "1999b" and "Witchas" should be "Witschas"
- Line 118: "requiring performing independent particulate…" is a bit awkward. Perhaps try, "requiring performance of independent particulate scattering…"
- Line 124: "implies…." could be replaced with "imposes a higher accuracy requirement on the laser pointing to maintain…"
- Figure 1: suggest: "receiver Telescope" (vs. "reception"), "boresight" vs. "steering"
- Line 240: suggest replacing "…we give here priory to.." with "…we give *priority* to…"
- 290: ODP should be OPD
- Line 367: "emitted the" should be "the emitted"
- Line 387: "…allied to a low reading noise." might be better as "…attributed to low read noise."

- Line 445:  "…in the due…" ? Should that be "…in the PBL, due…"
- Line 542:  "Witchas" should be "Witschas".
- Line 613-614:  This sentence is confusing. Do the authors mean "…can be derived from meteorological analysis to provide products with the required accuracy."?
- Line 651:   "celerity" is an unusual term for the usual "speed" of light terminology– but I guess it does apply here.
- Line 728:  There is no Witschas 2008.  Should that be 2010?

---

## Author Comment (AC1) · 25 Feb 2021

Answers to Reviewer 1

*1. An article by Stoffelen et al. (Bull. Amer. Meteor. Soc., 101, 2020) on the requirements and capabilities of future satellite-borne wind observations was recently published. As the study, amongst others, discusses the way forward toward an operational wind profiling mission, its conclusions should also be considered and referenced in the present manuscript.*

This reference will be included in the revised version of the manuscript and cited in the introduction.

*2. In the entire paper and especially in the introduction, I am also missing a reference to the Optical Autocovariance Wind Lidar (OAWL) developed by Ball Aerospace (Tucker et al., J. Atmos. Ocean. Tech., 35, 2018), although the citation is included in the reference list. The OAWL instrument also relies on a quadrature Mach–Zehnder interferometer and thus represents a very similar concept as the one described in the manuscript. Therefore, I suggest expounding on the advantages and disadvantages of the proposed lidar design compared to the OAWL system.*

The references will be added in the introduction. It is to be noticed that a reference on OAWL was cited in the text along with BP03 when the four-channel detection technique is introduced in section 3.2., this will be updated.

We believe that our all-prism design is, a priori, more stable than the free-space cat-eye design which depends on the mechanical stability of independent optical elements. Nevertheless it is difficult go further in this discussion without a careful comparison of the two systems which would be too long and out of the scope of our paper. The main differences between the OAWL and HSRD-LNG designs are discussed in section 3.2.

*3. Page 5, line 152: Please clarify what is meant by off-axis MI interferometer. Otherwise many readers might be lost in the following section.*

Better than trying an explanation that may be long and awkward, we will refer to Liu et al 2012 to introduce the off-axis (or tilted) MI.

*4. Page 7, lines 202 ff.:*

*- The proposed design envisages the use of multimode fibers for guiding the signal and reference beams to the receiver. What is the expected signal loss introduced by the fibers? Do the authors expect speckle noise to be an issue for the measurement accuracy?*

In the HSRD-LNG system, the transmission of the fiber arrangement, including the input and output lenses and the mode scrambler, is approximately 80% (added in section 3.1). The speckle noise is not an issue for the measurement accuracy, as the design is based on a large aperture and wide field of view. The receiver the number of spatial speckles Ms, defined as the ratio of the solid angle of the transmitted beam to the diffraction solid angle of the telescope is about $2.75 \ 10^4$. The number of temporal speckles Mt is around 500, due to the extended range gates. The total number of speckles M=Ms.Mt is then approximately $1.4 \ 10^7$. As the number N of collected photo-electrons per shot and range gate is on the order of $10^3$, the speckle noise (given by the ratio N/M) is much smaller than 1, so that the noise variance is not modified. Note also that the fiber beam étendue is necessarily larger than the telescope beam étendue in order to accept the received beam with a substantial alignment margin. The maximum number of spatial speckle of the fiber ($9 \ 10^4$) is then larger than Ms and does not limit the total number of speckle of the signal. This will be added in section 4.1.

- *How is the mode scrambler realized and what is its expected transmission based on the experience with the LNG?*

The beam scrambler principle is based on the fact that the near field (illumination distribution) at the fiber output, after a propagation of about 1m, is always uniform, whatever the light illumination at input (in the limits of the fiber beam acceptance). The mode scrambler arrangement consists in two lenses, the first lens images the output of the first fiber on the second lens while the second lens images the first lens aperture (not homogenously illuminated) on the second fiber. This way, the near and far fields at the output of the second fiber are uniform and fill the whole fiber beam étendue.  A plate with an uncoated face is also introduced in the arrangement in order to inject the reference signal with the same characteristics thanks to a symmetrical optical imaging (this will be added in section 3.1).

5. *Although the theoretical error estimation yields convincing results, it would be desirable if the authors could add some experimental results from the LNG to support their theoretical findings. Is it possible to transfer the results from Bruneau and Pelon (2015), particularly in terms of the SNR and the impact of solar background, to the proposed spaceborne lidar in order to estimate the systematic and random wind and backscatter ratio errors based on actual atmospheric measurements? In the discussion chapter, the authors state that "a better overall performance is expected to be achieved as based on realistic parameters derived from airborne operation with a minimized risk and increased design compactness and reliability". At the end of the conclusion chapter, it says "The performance analysis on the QMZ interferometer is supported by measurements performed in the frame of the UV HSRD-LNG airborne lidar developed and operated by LATMOS." Could they please elaborate on these statements?*

We agree with the reviewer that confronting observations and numerical simulations of measurements with our airborne lidar would be of added value to support our design proposal. We thus propose to add a new Appendix (Appendix C) to the paper, to show and discuss dispersion of measurements (bias and accuracy). As our system is performing nadir viewing, we propose to use slant observations obtained as the aircraft was turning. This allows to compare lidar VHLOS measurements through VAD analysis to collocated radiosonde measurements and analyze bias in retrievals as well as fluctuations. The Appendix would include figures based on dispersion analysis between VHLOS observations and VAD fit. The differences between lidar measurements and theoretical error calculation will be derived from our simulation model (from Eqs 6 and A17) to show statistical agreement. A table can be added to synthetize results.

6. *The quality of the figures should be improved. In particular, the resolution of Figs. 3 to 11 is rather low which makes it difficult for the reader to get the details of the plots.*
The quality of these figures will be improved.

Technical corrections listed by the reviewer will be made in the text.

---

## Author Response (AR1)

**Answers to Reviewers**

**Answers to Reviewer 1**

1. An article by Stoffelen et al. (Bull. Amer. Meteor. Soc., 101, 2020) on the requirements and capabilities of future satellite-borne wind observations was recently published. As the study, amongst others, discusses the way forward toward an operational wind profiling mission, its conclusions should also be considered and referenced in the present manuscript.

*This reference is now included in the revised version of the manuscript and cited in the introduction. Other references are also added to this respect (Baker et al., 1995, 2014, Stoffelen et al. 2005), or further in the text for radiation objectives (Stephens et al., 2002, Illingworth et al., 2015) in the text in coherence with the reference list (corrected).*

2. In the entire paper and especially in the introduction, I am also missing a reference to the Optical Autocovariance Wind Lidar (OAWL) developed by Ball Aerospace (Tucker et al., J. Atmos. Ocean. Tech., 35, 2018), although the citation is included in the reference list. The OAWL instrument also relies on a quadrature Mach–Zehnder interferometer and thus represents a very similar concept as the one described in the manuscript. Therefore, I suggest expounding on the advantages and disadvantages of the proposed lidar design compared to the OAWL system.

*References on OAWL development and results (Tucker et al., 2018, 2019, 2020) are added in the introduction. Coherence with reference list has been checked.*

*The all-prism design a priori appears more stable than the free-space cat-eye design which depends on the mechanical stability of independent optical elements. Nevertheless it is difficult go further in this discussion without a careful comparison of the two systems which would be too long and out of the scope of our paper. The main differences between the OAWL and HSRD-LNG designs are discussed in section 3.2.*

3. Page 5, line 152: Please clarify what is meant by off-axis MI interferometer. Otherwise many readers might be lost in the following section.

*Better than trying an explanation that may be long and awkward, we refer to Liu et al 2012 to introduce the off-axis (or tilted) MI.*

4. Page 7, lines 202 ff.:

    - The proposed design envisages the use of multimode fibers for guiding the signal and reference beams to the receiver. What is the expected signal loss introduced by the fibers? Do the authors expect speckle noise to be an issue for the measurement accuracy?

*Additional texts are added in section 3.1 to give more information. Section 3.1 is modified as*

*An optical mode scrambler is inserted on the fiber path in order to obtain a uniform illumination distribution at the interferometer input independently of the illumination distribution in the telescope focal plane (field stop). The mode scrambler consists in two lenses, the first lens images the output of the first fiber on the second lens while the second lens images the first lens aperture on the second fiber. This way, the near and far fields at the output of the second fiber are uniform and fill the whole fiber beam étendue. This arrangement ensures that the interferometer response is not biased by transmitter/receiver misalignment. A small plate and a symmetrical lens arrangement inserted in the mode scrambler allows the injection of a small amount of the emitted pulse used as*

the reference signal and transported by a second optical fiber. The fiber and the scrambler ensure the complete depolarization of the signal before it arrives on the MZI (even when including a polarization splitter in the front optics). This was implemented and successfully tested on the airborne LNG system. The overall efficiency of the fibers and mode scrambler on the atmospheric signal path is around 80%. The output of the fiber is then collimated by a 15-mm-focal-length lens at the MZI input port.

*and in section 4.1*
The speckle noise is not an issue for the measurement accuracy, as the design is based on a large aperture and wide field of view. The receiver number of spatial speckles Ms, defined as the ratio of the solid angle of the transmitted beam to the diffraction solid angle of the telescope is $2.75 \ 10^4$. The number of temporal speckles Mt is around 500, due to the extended range gates. The total number of speckles Mspec=Ms.Mt is then approximately $1.4 \ 10^7$. As the number N of collected photo-electrons per shot and range gate is on the order of $10^3$, the speckle noise (given by the ratio N/Mspec) is much smaller than 1, so that the noise variance is not modified. Note also that the fiber beam étendue is necessarily larger than the telescope beam étendue in order to accept the received beam with a substantial alignment margin. The maximum number of spatial speckle of the fiber ($9.10^4$) is then larger than Ms and does not limit the total number of speckle of the signal.

- How is the mode scrambler realized and what is its expected transmission based on the experience with the LNG?

*The beam scrambler principle is based on the fact that the near field (illumination distribution) at the fiber output, after a propagation of about 1m, is always uniform, whatever the light illumination at input (in the limits of the fiber beam acceptance). The mode scrambler arrangement is now detailed in the modified section 3.1 (see above).*

5. Although the theoretical error estimation yields convincing results, it would be desirable if the authors could add some experimental results from the LNG to support their theoretical findings. Is it possible to transfer the results from Bruneau and Pelon (2015), particularly in terms of the SNR and the impact of solar background, to the proposed spaceborne lidar in order to estimate the systematic and random wind and backscatter ratio errors based on actual atmospheric measurements? In the discussion chapter, the authors state that "a better overall performance is expected to be achieved as based on realistic parameters derived from airborne operation with a minimized risk and increased design compactness and reliability". At the end of the conclusion chapter, it says "The performance analysis on the QMZ interferometer is supported by measurements performed in the frame of the UV HSRD-LNG airborne lidar developed and operated by LATMOS." Could they please elaborate on these statements?

*We agree with the reviewer that confronting observations and numerical simulations of measurements with our airborne lidar would be of added value to support our design proposal. We thus added a new Appendix (Appendix C. Analysis of observed HSRD-LNG performance) to the paper, to show and discuss dispersion of measurements (bias and accuracy). We used slant observations obtained as the aircraft was turning. This allows to compare lidar VHLOS measurements through VAD analysis to collocated radiosonde measurements and analyze bias in retrievals as well as fluctuations. The Appendix includes additional figures (profiles, histograms of differences) based on dispersion analysis between VHLOS observations and VAD fit. The differences between lidar measurements and theoretical error calculation are derived from our simulation model (from Eqs 6 and A17) to show*

*statistical agreement for both wind and backscatter coefficient retrievals (X-Y plots). A table is added to synthetize results on wind measurements.*

6. The quality of the figures should be improved. In particular, the resolution of Figs. 3 to 11 is rather low which makes it difficult for the reader to get the details of the plots.
*The quality of these figures has been improved using bold fonts.*

*Technical corrections listed by the reviewer are made in the text.*

**\*\*\*\*\*\*\*\*\*\*\*\*\***

**Answers to Reviewer 2**

The paper has some confusing wording and sentence structure and would benefit greatly from editing by a native English speaker.
*We adopted English edits proposed by the reviewer and did our best to improve wording and sentence structure.*

Most of the figures, at least in the version this reviewer was provided, have font that is too small or blurry to read (especially figures 3-11). Please replace with figures with higher resolution and/or larger font.
*We now use bold fonts*

The paper presents a somewhat limited view of work done in the fields of wind lidar and HSRL. There is similar work being done in this field that the authors could use to further support and even enhance the important case that they are making in this paper. For example, presentations on OAWL at the Aeolus CalVal workshop have also supported the idea of using a short optical path difference QMZ interferometer for Aeolus and have demonstrated comparison with Aeolus data. While the interferometer implementation is different, the mathematical concept and listed advantages are quite similar.
*Although a reference to OAWL was already included in the reference list, the text corresponding to the reference was partly removed. References to the new results (which provide additional support to our paper, as mentioned by the reviewer) as reported at the Aeolus Cal/Val workshop were added.*

A number of variables in the text (including italic font, subscripts, symbols) do not match those in the equations. This will presumably be caught in a more thorough editing process. A few of the publications listed in the references section appear to be missing in the text: Baker, Benedetti, Lux, Reitebuch 2020, and Tucker. (Note, the text refers to Reitebuch 2019a, and Reitebuch 2019, but there is only one Reitebuch 2019 in the list of references. Perhaps one should be the Reitebuch 2020 reference?)
*The text and reference list have been edited and we have added reference to Baker et al., 1995 and 2014, Stephens et al., 2002, Illingworth et al., 2015 in the text.*

• Line 14: Might reword to say, "...wind profiler using a *single* fixed line-of-sight lidar from space." As some other proposed concepts have looked at fixed dual lines-of-sight.
*OK*

• Line 19: "This ability is…" To what ability is the sentencing referring?
*To make it clearer, the sentence is rephrased as*
*"This ability to profile wind and cloud-aerosols radiative properties is meeting"*

• Line 30: could clarify to say that, "… backscatter coefficients can be retrieved with uncertainties better than a few percent where backscatter levels exceed XX%, such as in the boundary layer and …."
*OK. We propose to refer to the scattering ratio more commonly used to define backscatter level. The new sentence is*
*"… better than a few percent when the scattering ratio exceeds 2, such as …"*

• Lines 65-80: This is a limited list of HSRL approaches and could be expanded to include the OAWL approach as well as work being done by Fahua Shen:
o Fahua Shen, Jie Ji, Chenbo Xie, Zhao Wang, and Bangxin Wang, "High-spectral-resolution Mie Doppler lidar based on a two-stage Fabry–Perot etalon for tropospheric wind and aerosol accurate measurement," Appl. Opt. 58, 2216-2225 (2019)
*The work of Shen et al. is referring to Fabry-Perot interferometric analysis but the goal of these lines is rather to focus on MZI. The proposed reference was thus not included but the other suggestions from reviewer 2 have been accounted for. These lines have been rewritten as:*
*"In the United States, Ball Aerospace developed a Doppler wind lidar including a spectral discriminator based on the same principle of Quad-channel MZI (Grund et al., 2009) which was successfully operated onboard the NASA WB57 aircraft (Tucker et al., 2018) and further involved in the Aeolus CA/VAL (Tucker et al., 2019, 2020)."*

o Tucker, S. and Weimer, C., 2018, October. Benefits of a quadrature Mach Zehnder interferometer as demonstrated in the Optical Autocovariance Wind and Lidar (OAWL) wind and aerosol measurements. In Remote Sensing of the Atmosphere, Clouds, and Precipitation VII (Vol. 10776, p. 107760E). International Society for Optics and Photonics.
*More recent references on OAWL will be added (see above)*

• Line 123: DE-FP and Fizeau interferometers will have a "small" acceptance angle based on the finesse. Low Finesse etalons will have higher acceptance angles than higher finesse ones, but will provide poor frequency discrimination. Perhaps this can be clarified in the sentence by adding something like…, "*If they are providing high frequency discrimination*, both DE-FP and Fizeau interferometers will have small angular acceptance."
*As the sentence is referring to the ALADIN context (and not a more general one), no modification is to be made*

• Line 125: Don't the narrow interferometer fields of view (FOVs) *also* impose a high accuracy requirement on the alignment between the telescope and the receiver?
*Yes agreed, the sentence is modified as*
*"… the large telescope aperture is resulting in a very small field-of-view and imposes a high accuracy requirement on the transmitter-receiver co-alignment."*

• Line 134: Could also include Grund 2008
o Grund, C.J., Howell, J., Pierce, R. and Stephens, M., 2009, April. Optical autocovariance direct

detection lidar for simultaneous wind, aerosol, and chemistry profiling from ground, air, and space platforms. In Advanced Environmental, Chemical, and Biological Sensing Technologies VI (Vol. 7312, p. 73120U). International Society for Optics and Photonics.
*The reference to Grund et al., 2009 has been added (see above)*

• Lines 149-153: The frequency stability is only required IF an accumulation detector is used for the observation. If a higher speed detector is used, such as a PMT or APD (e.g., CALIPSO detectors) then the reference can be updated every pulse, if needed.
*Agreed. The sentence is modified as*
*"Appropriate frequency stability is just required during the signal accumulation needed for a single measurement"*

• Line 161: 1.4 or sqrt(2)? Is this just for simplicity?
*It is sqrt(2), but 1.4 reads better and it is thus kept as a decimal number. An explanation has been added at the end of the* sentence as
*"… higher optical efficiency (factor 2)."*

• Line 164-168: This may be due to the grammatical structure of the sentence, but it seems that the authors are implying that (because of the reflection from one edge of the Aeolus ALADIN DEFP is being used in the other edge) the Aeolus approach has a theoretical 1.4x advantage over the QMZI approach. However, it's unclear how it is possible to provide perfect efficiency through a double edge approach while maintaining the necessary frequency discrimination. Unlike with the QMZ approach, there will always be some molecular backscatter signal that is not allowed to pass through either edge filter and thus will be lost. I am aware the authors are quite familiar with how Aeolus (and DEFP) operates, so perhaps this was not the intent of the paragraph, but it's important to not lead other readers into an unintended comparison.
*This is just a factor 2 in energy (split or combined beams), as above. Modified as*
*"… larger than 1.4 (limit case of a factor 2 gain in the transmitted energy)"*

• Line 176-179: These statements could use a little bit of qualification. For example, *for short OPDs*, the particulate signal will be near unity, but not if the OPD is long compared to the laser linewidth. Perhaps just state that the discussion refers to the 3.2 cm OPD concept being presented here.
*Yes, rephrased as*
*"Provided the emission linewidth is sufficiently narrow, the particulate signal produces an interference contrast near unity, significantly higher than that produced by the molecular signal (see appendix A, Eq. A14). "*

• Line 189: The term "emitter" is used here (and elsewhere in the paper, including Table 2) however the term "transmitter" is used in much of the Aeolus literature (e.g., Reitebuch, et al., 2009).
*All the text has been modified to change "emitter" into "transmitter"*

• Line 190: This point about "acceptance angle" (or "maximum field angle" or "range of field angles") is an important point to make!
*Yes indeed*

• Line 195: Suggest the wording "…can be mounted on the same plate, *as was done on CALIPSO*, and *boresight* mirrors…." (such language indicates heritage for the design)
*Agreed. Done*

• Line 220: perhaps add: "..to obtain a uniform illumination *and uniform field* distribution…"
*Rewritten as*
*"uniform illumination at the interferometer input independently of the illumination distribution in the telescope focal plane (field stop)."*

• Line 225: Also successfully implemented and tested on the airborne OAWL system (which further supports your case!)
*OK*

• Line 265: Do you mean that each spot fills an 8x8 pixel area?
*Yes, rewritten to make it more explicit as*
*"Each spot fills a 8 by 8 pixels area in the …"*

• Line 261-270: The discussion is focused on using the same A-CCD as was used on Aeolus, however discussions of using a new ACCD or different type of detector are under consideration for Aeolus. Could the authors suggest what approach would be idea for a QMZ design? Perhaps ranging from more ACCD rows, to a different type of detector?
*The detectors are a critical part of the system in terms of intrinsic noise, this is the first parameter to consider, but various technical solutions are possible. This discussion is just intended to focus on this question without going into technical details.*

• Table 2: "Emitter Linewidth": according to the text, the 200 MHz represents the allowed spectral jitter or drift over the 50 pulse (1/2 second) accumulation time and likely does not represent the laser linewidth. Perhaps clarify this by calling it "Accumulated Emitter Linewidth (1/2 s)" or even "Allowed laser linewidth". Also, (see above comment on "emitter" vs. "transmitter").
*OK. Text clarified as* Accumulated Emitter Linewidth

• Line 293: Clarify that "*For a 3.2 cm OPD*, the modulation by the molecular return…." For some longer OPDs, molecular return will provide NO modulation, just offset.
*Modified accordingly*

• Line 305: YES, this is true, however won't there be a backscatter ratio below which the contrast difference is too small to clearly estimate?
*The random error on $V_{LOS}$ is inversely proportional to $M_{atm}$ (Eq. A16) which is itself given by Eq. A4. The error dependence with the backscatter ratio can be easily derived from a combination of these two equations, all things equal otherwise.*

• Line 330: Be sure to clarify that because of the *specific type* of field compensation used in this specific design, the OPD variation is only dependent on 4th power of the source angle. There are other types of field compensation (e.g. cat's eye) for which the OPD variation
*The text is clarified as*

"Note also that, for our on-axis field compensated design, the variation of OPD is only dependent on the fourth power of the source angle (the second power dependence is canceled and the third order dependence is null due to all on-axis optical elements). "

• Line 331: Please provide a reference for the equation provided in this line.
*Instead of giving a reference, the calculation of the residual wavefront error given by the beam étendue is detailed in Appendix B (now renamed as* MZI field compensation and aperture*.)*

• Line 340: Indeed, smaller incident angles on beamsplitters make it easier to balance reflection and transmission values.
*Yes, but other considerations such as beam path separation in the interferometer require to increase the incidence angle to about 40-45°.*

• Line 355: The authors suggest using the laser in multimode operation to perform the amplitude (sensitivity parameter) calibration– but for many seeded lasers, running them multi-mode comes with a risk of damage due to modal interference.
*Yes, but the laser should be designed to withstand this multi-mode operation in case of seed laser failure.*

• Line 337: modulation calibration can also be performed by allowing the laser to drift in frequency while remaining seeded. This can be achieved fairly easily by temperature tuning the seed laser source, or laser cavity.
*Agreed. Added part to the sentence:*
"or by tuning the laser seeder frequency"

• Line 369-374: Please indicate the wavelength for these profiles in the text for clarity. Have these profiles been validated in any sense using data from Aeolus and/or CALIPSO?
*355nm information added*

• Figure 4: Please increase the text font size and overall figure clarity. The text is too blurry to read. Sub figures (a) and (b) do not have titles indicating the wavelength, etc. Likewise, please indicate the wavelength for these profiles in the figure caption.
*OK, improved.*

• Section 4.2: It's nice to see this study on the impact of the horizontal light of sight angle on performance, though it would be interesting to compare it to previous studies done for Aeolus back when the mission was selected. This type of angle analysis shouldn't be unique to this particular design, but if it is (for some reason this reviewer is missing) then please clarify. It indicates that the difference in 1/R^2 is less than the error in retrieving the horizontal LOS from a smaller pointing angle.
*We don't have information on reasons of the choice of a 35° angle for Aeolus. But clearly, our calculations show that a slant angle around 45° gives a smaller $V_{HLOS}$ error.*

• Table 3: The authors may wish to clarify here that the total power is the same for the two systems. Aeolus requirements were based on a 2x higher laser pulse energy, and the QMZ approach is based on the as-operating Aeolus pulse energy but with a 2x pulse repetition frequency.

*Mentioned in the text.*

• Line 481: "where such an accuracy may be more acceptable than bias for assimilation purpose…" The terms "accuracy" and "bias" are often tied together. Do the authors mean that precision (or "uncertainty" or "random error") is more acceptable than bias?
*Clarified (precision as random error)*

• Line 508: The authors refer to a 10-3 relative error in the sensitivity calibration of each QMZ channel, however it might be good to clarify that this does (or does not) include the detector response sensitivity. For example, if one channel sees a high portion of the interferometric fringe during the outgoing pulse, will the detector response have any impact on returns for that channel?
*Yes, the calibration of parameters $a_i$ clearly includes the detector. Our calculations give the sensitivity of the measurement bias with regard to these parameters. If the linearity of the detector is an issue, this calibration should be done at different illumination levels and corrections should be applied accordingly. However, we will not discuss this possibility because it depends on the characteristics of the detectors, on which we have few details.*

• Line 564: the authors should clarify that "…the *short OPD* QMZ does not attempt to separate molecular…." There are long OPD QMZI designs that easily separate these two.
*Yes, but the paper focuses on the use of a single MZI, and this turns to the use of a short OPD. Long OPD MZIs are discussed in BP03 (referred to in the text before). The sentence is modified as " …the proposed QMZ …"*

• Line 619-620: Is it possible to show a scaling of the performance of the UV HSRD-LNG airborne lidar to space-based operation (e.g., see Baidar et al. 2018).
• Appendix A: This is a good review of previous papers by the authors describing QMZ performance. Has data from the LNG system been used to validate this model and if so, how did they compare?
*We added a new Appendix (Appendix C Analysis of observed HSRD-LNG performance) to present results obtained with LNG and discuss bias and errors as compared to simulations.*
*The simulation model has been applied to LNG observations. Sentences are added in the text to refer to this addition. More particularly at the end of section 4.1, it is added that*
*"Appendix C shows measurements performed with HSRD-LNG where the error calculated with equations similar to that used in the performance model is compared to the experimental error. Fig. C1 shows that the calculated and experimental errors are in good agreement. A comparison between lidar wind measurements, including calculated error, and collocated radiosonde profiling is also presented in Appendix C (Fig. C2 and Table C1). A similar comparison of measured and calculated errors on the backscatter coefficient shows also a good agreement (Fig. C3). We can therefore consider that these results validate the equations used in the performance model."*

• Line 670-684: Why is 3 cm being used here for the OPD vs. the 3.2 cm chosen for the optimal system?
*Though 3.2 cm is the final choice for the OPD, in this section we present only preliminary calculations of $M_{par}$ and $M_{mol}$ and a value of 3 cm is indicative for the range of OPDs considered in the paper.*

• Line 689: The assumption that the background can be measured over a long duration may meet challenges with highly varying cloud albedo over orbit.

*Long duration stands for a long range gate as compared to the range gate required for vertical resolution. Background and signal measurements can be separated by less than 0.001s that corresponds to a separation of a few meters along track.*

• Appendix B: It's quite nice to see this analysis in a publication, showing the impact of OPD and maximum aperture driven by étendue and it would be nice to see this in the main paper, if room allows. It's important to clarify, however, that the analysis applies to QMZ structures using field compensation plates.

*We made the choice to keep the main part as focused as possible. The field compensation is now more detailed in the new Appendix B (including 7 more equations).*

• Appendix C: Many of the points made in this section are really important in the argument for using a QMZ type system for an Aeolus follow-on. If possible, can more of this be moved into the main part of the document?

*Yes, only a small part of it is in the text (sections 3.3, 4.4 for example). As mentioned, we want to keep the main text as focused as possible. We think that interested readers will read the new Appendix D (*Atmosphere-modelling dependent errors in the retrieval of particulate backscatter and extinction coefficients*).*

*Modifications have been made accordingly to detailed comments*

---

## Referee Report (RR1)

**A new lidar design for operational atmospheric wind and cloud aerosol survey from space**

**By Didier Bruneau & Jacques Pelon**

**Summary and General Assessment**

This paper builds on the decades of work performed and published by the authors (and their colleagues) on using Quadrature Mach Zehnder interferometry for wind and aerosol lidar systems. The paper represents an update on the QMZI approach and how it may provide an alternative to the Double Edge Fabry Perot an Fizeau spectrometers used on Aeolus, to simplify the mission, potentially without making many changes in other areas. Using a radiometric model and interferometer receiver model, the authors demonstrate optimization of the system for full tropospheric coverage, resulting on a QMZ design with 3.2 cm OPD to optimize fringe contrast (and thus optimize wind retrievals) from molecular backscatter. The modeling approach is validated using a slightly different design on demonstrated on an airborne platform.

The paper has several appendices, making for an extensive paper, however much of the content has been presented in previous publications and is provided for ease of reference. The new section on comparing the HSRD-LNG system to performance studies is important, though there are some new questions regarding the comparisons (see detailed comments).

The paper points out important advantages of the QMZ approach (e.g., a reduced need for frequency stability, high efficiency/optical throughput, and high angular acceptance/field of view) and its potential value for future wind mission systems. With minor edits, as listed below, I recommend this paper for publication in AMT.

**General Editorial Suggestions**

The figures have been improved in this version and include legible text, and references appear to have been fixed. The paper does still contain some confusing wording and sentence structure that would benefit from editing by a native-English speaker. In case this is not feasible, and because AMT will not provide such editing, some additional suggestions have been provided here (but no effort is made to maintain the symbols/subscripts).

**Technical Comments**

- Line 75: Is this really the first airborne HSRL? Sroga, Eloranta, et. al., 1983 -  (airborne HSRL demonstration in 1980). Also see Shipley et al. 1983. Also important to recognize the work of Eloranta et. al with NCAR –the (HIAPER) GV-HSRL system.

- Line 77-84 – this addition of more background information on the LNG is quite helpful, especially on connection with the new Appendix C.

- Lines 270-274:  The cat-eye arrangement described in Tucker et al. 2018 (based on Wang et al. 2000) provides an 8 mrad FOV for a QMZ with a 90 cm OPD (challenging for a solid MZI) ; and the description indicates it does not require delicate mechanical stability for the same reasons that QMZs do not require delicate laser frequency stability if one is capturing the reference phase.

  Suggest replacing:

- o "An alternative design of a field-compensated MZI is given by the cat-eye arrangement (Tucker et al., 2018). This fully reflective design eliminates the light path through the glass which can cause a wavefront distortion but leads to a bulkier arrangement which requires a delicate mechanical stability. Additionally, as discussed in section 3.4, an all-prism MZI design with a small OPD can achieve high quality and the thermally induced wavefront distortion can be easily controlled."

  with

- o "An alternative approach to field-compensating an MZI is to use cat-eye arrangement as demonstrated for a 90 cm OPD in Tucker et al., 2018.**"**

- Line 412-419: While this new discussion on speckle is highly useful, it appears to interrupt the discussion on sampling and accumulation. Perhaps move it to just before the line starting with "Appendix C shows measurements performed with…."

- Line 423: Regarding this sentence "The signals are summed on 14 elementary samples corresponding to a total of Ns=700 shots for an observation horizontal resolution of 50 km" –

  - o What is an "elementary sample" – this appears to be the first mention of the term. Some readers might understand that it refers to a 50 shot ACCD accumulation, but please clarify.

  - o Assuming that elementary samples refers to sets of 50 shots (0.5s accumulations at 100Hz PRF), then if the signals are summed over that time (14 x 0.5s = 7s) does the laser frequency actually need to be stable over 700 shots (7 seconds)?

  - o Perhaps the word "summed" could be replaced with "reference phase adjusted and accumulated" ?

- Figure 4: While two of the figure subplot titles now list the wavelength in their title, it would still be helpful to indicate the wavelength (for all the profiles and subplots) in the figure caption.

- Lines 543-545: The second half of the sentence is unclear: "… but in regions where the aerosol load is significant ($R_{\mathbb{G}} \geq 2$) the contributions of the uncertainty on βmol and Mmol are of the same order of magnitude" The contributions of which uncertainty on Bmol and Mmol are of the same order of magnitude? Do the authors mean to say "…but in regions where the aerosol load is significant ($R_{\mathbb{G}} \geq 2$), contributions to the total error from uncertainty in βmol and Mmol are of the same order of magnitude."

- Appendix C: It is important that that authors have added a new section to show validation of the instrument performance model.

  - o Regarding Figure C1b - Is the standard deviation (sigma) estimate (provided in equation A-17) a minimum sigma, or a lower bound on sigma? If so, how is it that the measured sigma is sometimes lower than the calculated sigma? Wouldn't one expect that with the added natural atmospheric variability, the measured standard deviation would always be larger than the model?

  - o Line 808: The equation A-17 used in the comparison shown in Figure C1b does not include "N" (number of shots, number of samples) used in the estimate. What parameters were used in A-17 for SNR, Mo, Matm, and how were the accumulations accounted for in the measurements vs. the model? This can probably be addressed with a few simple parameters including SNR, Mo, and Matm used in A-17, as well as a 1/sqrt(N) factor to account for the various pulses used in the estimates.

- o If the authors are short on room, the authors could remove lines 814-833 and just reference the Bruneau 2015 (or 2020 conference paper "Operation of the airborne 355 nm high spectral resolution and doppler lidar LNG) instead. Unless perhaps other reviewers have requested validation of the general QMZ approach It's not clear that the newly added radiosonde comparisons for the airborne campaign are critical to the main paper.
- o Figure C3: like for Figure C1b, it's unclear how the measured standard deviation can be smaller than the ideal calculated standard deviation unless the calculation is itself based on an uncertain variable (e.g. SNR?). Which parameters

**General English editing/grammatical suggestions**

- Lines 19-21 – still some grammatical confusion in the abstract, suggest the following for this sentence: "This ability to profile wind and cloud/aerosol radiative properties enables meeting the two highest priorities of the meteorological forecasting community regarding atmospheric dynamics and radiation."

- Line 22: suggest

  "We discuss the optimization of the key parameters necessary in the selection of a high performance system…"

- Line 31: suggest

  "The chosen design further allows addition of a dedicated channel for aerosol and cloud polarization analysis."

- Line 96: Perhaps replace "take over" with "extend and improve upon" (if indeed it will improve)

- Line 103: remove the comma after "2", "in section 2 we…"

- Line 120: replace "reading noise" with "read noise"

- Line 121-122: suggest replacing

  - o "but the interferometer configuration has revealed operation and performance constraints proper to the choice made"
    with

  - o "but operation has revealed performance constraints set by the Rayleigh channel fabry perot interferometer designs and configurations."

- Line 170: Remove "besides"

- Table 2 caption:

  - o Suggest replacing "Parameters are compared to Aeolus ones as a reference actual in space values reported by Reitebuch et al., 2019"

  - o With "Parameters for the proposed system as compared to those from the on-orbit Aeolus, as reported by Reitebuch et al., 2019."

- Line 235: Replace "consists in two lenses,…" with "consists of two lenses,.."

- Line 285: Replace "a 8x8 pixels area" with "an 8x8 pixel area"

- Line 338: Replace "…leads for the chosen parameters" with "…leads to the chosen parameters"

- Line 409: Replace "reading noise" with "read noise"

- Line 410:  Replace "that add" with "that adds"

- Line 442:  Replace "similar to that used" with "similar to those used" as it refers to multiple equations used.

- Line 521:  suggest:  "The particle backscatter and extinction retrieval (Eq. A8 –A10) requires  knowledge of $\beta_{mol}$ and $M_{mol}$."

- Line 538:  suggest replacing

  - "Note that the wind speed variation or turbulence in the probed volume can induce a variation in $M_{par}$. With a Doppler shift of 5.6 MHz per ms-1 the broadening of the particle backscattered spectrum is very small as compared to an emitted linewidth of 100-200 MHz and the incidence on $M_{par}$ is negligible. We can then consider that $M_{par}$ is not affected by the atmospheric conditions"

    With

  - " Note that wind speed variation or turbulence in the probed volume could slightly reduce $M_{par}$, however with a Doppler shift of 5.6 MHz per ms-1 the broadening of the particle backscattered spectrum is much less than the emitted laser linewidth of 100-200 MHz, and thus the impact on $M_{par}$ is negligible. We therefore consider that $M_{par}$ is not affected by the atmospheric conditions."

- Line 547:  suggest replacing "quadratically added to…" with the common term "root square summed (RSS) with…"

- Line 621:  suggest replacing  "Unlike usual HSRL…" with "Unlike the usual filter-based HSRL…"

- Line 647:  replace "similarly to" with "similar to"

- Line 670-671:  replace "meteorological analyzes" with "meteorological analyses"  (analyses being the plural of analysis).

- Line 790: suggest replacing

  - "With our instrumental parameters we have finally a the residual wavefront distortion $\sigma_{RMS}$ = 3.7 10-2 nm due to the field compensation"

    with

  - "With our instrument parameters we find a residual wavefront distortion of $\sigma_{RMS}$ = 3.7 10-2 nm due to field compensation."

- Line 800:  suggest replacing,

  - "allowing to obtain several LOS measurements at various levels in the cloud, 800 allowing to perform a velocity azimuth display …"

    with

  - "providing several LOS wind measurements at various levels in the cloud, allowing performance of a velocity azimuth display…"

- Line 855:  should "6.10-3" be "6x10-3"?

---

## Author Response (AR2)

**Answers to Referees, 2nd review.**

**Reviewer 1**

The authors have addressed all the points raised in my review and revised the manuscript accordingly. Therefore, I recommend publication of the manuscript after the following two minor technical corrections:

1. For the citation of webpages in the introduction (l. 65, l. 118, l. 138, l. 245) a representative title should be used instead of the URL. The latter is supposed to be included in the reference list together with corresponding title and access date.

These references are now identified as (Aeolus-ESA-Portal-ext, date) with "ext" defined as the two extensions "forecast" and "mission", respectively

2. In Table 2, third line from top, 7.2 km/s should be changed to 7.2 km·s-1 in order to be compliant with the journal's formatting style.

Identified modification in Table 2 has been made.

**Reviewer 2**

• Line 75: Is this really the first airborne HSRL? Sroga, Eloranta, et. al., 1983 - (airborne HSRL demonstration in 1980). Also see Shipley et al. 1983. Also important to recognize the work of Eloranta et. al with NCAR –the (HIAPER) GV-HSRL system.

Although we had the Shipley et al. paper high in our tablets, we omitted the airborne aspect of this development. The work of Ed Eloranta was also more associated with ground-based measurements. Thank you to have refreshed our memories, and our apologies to these authors. We have modified the sentence as

After the first pioneering developments using Fabry-Perot interferometers (Shipley et al, 1983; Sroga et al, 1983), the first operational airborne HSRL systems were developed in the USA at U. Wisconsin [Eloranta et al., 2008] and at NASA [Hair et al., 2008], as well as in Europe at DLR [Esselborn et al., 2008], all these systems being based on the Iodine cell absorption technique.

**And added three more references**

Shipley, S. T., D. H. Tracy, E. W. Eloranta, J. T. Trauger, J. T. Sroga, F. L. Roesler, and J. A. Weinman, "High spectral resolution lidar to measure optical scattering properties of atmospheric aerosols. 1: Theory and instrumentation," Appl. Opt. **22**, 3716-3724, 1983. Sroga, J. T., E. W. Eloranta, S. T. Shipley, F. L. Roesler, and P. J. Tryon, "High spectral resolution lidar to measure optical scattering properties of atmospheric aerosols. 2: Calibration and data analysis," Appl. Opt. **22**, 3725-3732, 1983.

Eloranta E. W., I. A. Razenkov, J. Hedrick and J. P. Garcia, "The Design and Construction of an Airborne High Spectral Resolution Lidar," *2008 IEEE Aerospace Conference*, pp. 1-6, doi: 10.1109/AERO.2008.4526390, 2008.

• Line 77-84 – this addition of more background information on the LNG is quite helpful, especially on connection with the new Appendix C.

**Thank you. We thought a brief historical information could be useful.**

• Lines 270-274: The cat-eye arrangement described in Tucker et al. 2018 (based on Wang et al. 2000) provides an 8 mrad FOV for a QMZ with a 90 cm OPD (challenging for a solid MZI); and the description indicates it does not require delicate mechanical stability for the same reasons that QMZs do not require delicate laser frequency stability if one is capturing the reference phase. Suggest replacing:

"An alternative design of a field-compensated MZI is given by the cat-eye arrangement (Tucker et al., 2018). This fully reflective design eliminates the light path through the glass which can cause a wavefront distortion but leads to a bulkier arrangement which requires a delicate mechanical stability. Additionally, as discussed in section 3.4, an all-prism MZI design with a small OPD can achieve high quality and the thermally induced wavefront distortion can be easily controlled."

with

"An alternative approach to field-compensating an MZI is to use cat-eye arrangement as demonstrated for a 90 cm OPD in Tucker et al., 2018."

We have been asked by Reviewer 1 to compare the advantages and disadvantages of the proposed design with OAWL. So we tried to give the main lines of this comparison, in a few words. We agree that a 90 cm OPD with an all-prism MZI is unrealistic, but we actually address Doppler measurements relying mainly on molecular scattering which require a much smaller OPD. We do not fully agree with the reviewer when he proposes to shorten the explanations on the selected designs, but we agree it should be a more balanced presentation. In our opinion, the main difficulty of a cat-eye design lies in its mechanical stability (even with a monitoring of the reference during operation) while the main disadvantage of the all-prism design lies in its thermal sensitivity. To be fair, and more explicit, we replace the two sentences by:

An alternative design of a field-compensated MZI is given by the cat-eye arrangement (Tucker et al., 2018). This fully reflective design eliminates the light path through the glass which can cause a wavefront distortion but leads to a bulkier arrangement which requires a high mechanical stability. On the opposite the all-prism MZI design is insensitive to vibrations but is sensitive to temperature gradients. Nevertheless, as discussed in section 3.4, for a small OPD the all-prism design can achieve high quality and the thermally induced wavefront distortion can be easily controlled.

• Line 412-419: While this new discussion on speckle is highly useful, it appears to interrupt the discussion on sampling and accumulation. Perhaps move it to just before the line starting with "Appendix C shows measurements performed with...."

Agreed. The sentence is moved in the newly revised version.

• Line 423: Regarding this sentence "The signals are summed on 14 elementary samples corresponding to a total of Ns=700 shots for an observation horizontal resolution of 50 km" –

 $\circ~$  What is an "elementary sample" – this appears to be the first mention of the term. Some readers might understand that it refers to a 50 shot ACCD accumulation, but please clarify.

 $\circ$  Assuming that elementary samples refers to sets of 50 shots (0.5s accumulations at 100Hz PRF), then if the signals are summed over that time (14 x 0.5s = 7s) does the laser frequency actually need to be stable over 700 shots (7 seconds)?

 $\circ~$  Perhaps the word "summed" could be replaced with "reference phase adjusted and accumulated"?

**To be more explicit, we added some information as**

The signals are first accumulated on the ACCD over 50 shots forming an elementary sample. Then 14 elementary samples are reference adjusted and accumulated, corresponding to a total of Ns = 700 shots, for an observation horizontal resolution of 50 km and the resulting total SNR is

$$SNR = \frac{S_{tot}\sqrt{N_s}}{\sqrt{S_{tot} + S_b}}$$

• Figure 4: While two of the figure subplot titles now list the wavelength in their title, it would still be helpful to indicate the wavelength (for all the profiles and subplots) in the figure caption.

• Lines 543-545: The second half of the sentence is unclear: "... but in regions where the aerosol load is significant ( $R \Leftrightarrow \geq 2$ ) the contributions of the uncertainty on  $\beta$ mol and Mmol are of the same order of magnitude" The contributions of which uncertainty on Bmol and Mmol are of the same order of magnitude? Do the authors mean to say "...but in regions where the aerosol load is significant ( $R \geq 2$ ), contributions to the total error from uncertainty in  $\beta$ mol and Mmol are of the same order of magnitude."

**OK, we propose to rewrite as**

In clear air the uncertainty on  $M_{mol}$  dominates the total error on the backscatter and extinction coefficients (see Appendix D, Eqs D1 and D2), but in regions where the aerosol load is significant  $(R_{\beta} \ge 2)$ , contributions to the total error from uncertainty in  $\beta_{mol}$  and  $M_{mol}$  remain small and comparable in magnitude.

• Appendix C: It is important that that authors have added a new section to show validation of the instrument performance model.

• Regarding Figure C1b - Is the standard deviation (sigma) estimate (provided in equation A-17) a minimum sigma, or a lower bound on sigma? If so, how is it that the measured sigma is sometimes lower than the calculated sigma? Wouldn't one expect that with the added natural atmospheric variability, the measured standard deviation would always be larger than the model?

Equation A-17 gives the theoretical standard deviation  $\sigma(V_{LOS})$  caused by detection noise. It is the standard deviation that would be obtained on an infinite series of measurements. In Fig. C1-b each point of the observed standard deviation is an estimate of this standard deviation obtained on only one hundred measurements. This standard deviation estimate is itself a stochastic variable. Its mean value would reach  $\sigma(V_{LOS})$  for an infinite averaging (in the absence of additional variability). It is thus not impossible to obtain standard deviation estimates lower than the theoretical standard deviation even if an additional cause of variability is brought by the atmosphere (as mentioned in the text, the average of the observed standard deviations is slightly higher than  $\sigma(V_{LOS})$ , due to this additional variability).

Line 808: The equation A-17 used in the comparison shown in Figure C1b does not include "N" (number of shots, number of samples) used in the estimate. What parameters were used in A-17 for SNR, Mo, Matm, and how were the accumulations accounted for in the measurements vs. the model? This can probably be addressed with a few simple parameters including SNR, Mo, and Matm used in A-17, as well as a 1/sqrt(N) factor to account for the various pulses used in the estimates.

We believe that the measurement conditions are clearly defined. Equation A-17 includes the detection SNR which itself depends on the number of accumulated shots N (100) and on the vertical resolution (100m). As discussed in section 3.4 the intrinsic modulation factor  $M_0$  of the HSRD-LNG is 0.65. Actually, the SNR is derived directly from the recorded signals, as well as the observed  $M_{atm}$ . These parameters, specific to the measurement conditions, are used in Eq. A-17 for the calculation of  $\sigma(V_{LOS})$ .

 If the authors are short on room, the authors could remove lines 814-833 and just reference the Bruneau 2015 (or 2020 conference paper "Operation of the airborne 355 nm high spectral resolution and doppler lidar LNG) instead. Unless perhaps other reviewers have requested validation of the general QMZ approach It's not clear that the newly added radiosonde comparisons for the airborne campaign are critical to the main paper.

The statistics resulting from the comparison between lidar and dropsonde measurements have not been presented previously. Measurement bias is an important parameter in the assimilation process, requiring potential corrections as it is now the case for Aeolus. The histograms of figure C2-d and table C1 show that wind measurements can be carried out by the QMZ technique with a very limited bias. These results are related to the validation of the statistical error (bias and precision), and we believe they are of great interest in the context of the paper to show that the measurement bias with a QMZ can be very low. • Figure C3: like for Figure C1b, it's unclear how the measured standard deviation can be smaller than the ideal calculated standard deviation unless the calculation is itself based on an uncertain variable (e.g. SNR?). Which parameters

Same answer as above.

All suggested modifications to improve English have been implemented in the text.

**Additional modifications**

Last modifications have been made the text.

In the appendix C, one should read LOS instead of HLOS (now modified as)

Taking the sine fit as reference, we can compute the standard deviation of the LOS measurements and compare it to the standard deviation calculated with Eq. A17. Figure C1b shows the comparison of the measured and calculated LOS wind speed standard deviations computed at different altitudes on three different VAD measurements taken at 16:37, 17:34 and 18:00 UTC (for a total of 98 measurements).

The reference to the work of Herbst and Vancken (2016) on UV Doppler lidar using a Michelson interferometer has been added in section 2 and in the reference list.

Herbst, J., Vrancken, P.: Design of a Monolithic Michelson interferometer for Fringe-Imaging in a Near-Field, UV, Direct Detection Doppler Wind Lidar, Applied Optics, 55, 25, 6910-6929, 2016.

---

## Author Response (AR3)

**Associate Editor Decision: Publish subject to technical corrections (01 May 2021) by Oliver Reitebuch**

**Comments to the Author:**

With this the manuscript is accepted. Please revise Table 2 for the actual satellite altitude of Aeolus with a mean altitude of 320 km, with an orbital variation of about +-15 km.

**Answer to the Editor**

*Thank you for the acceptation of the paper and for pointing this correction. The altitude reported in Table 2 has been modified adding a second line for the altitude as "320 km (in space)"*

*As the text in section 5 needed to be modified, we changed lines 598 to 599, accordingly. Starting on line 595 this reads*

Aeolus has been studied to be operated with a 130 mJ / 100 Hz transmitter to meet the requirements but the development of the operational system only allowed using a 65 mJ / 50 Hz transmitter. Considering equations of errors with SNR depending on the square root of the emitted power (product of energy and repetition rate) at high SNR, the reduction in the horizontal resolution (90 km instead of 50 km), the change in altitude (320 km instead of 395/425 km), ALADIN in space should produce a statistical error increased by a factor of about 3.5 as compared to its original dimensioning.